# Joint Inference for Neural Network Depth and Dropout Regularization

**Kishan K C**[1]    **Rui Li**[1*]    **Mahdi Gilany**[2]
[1]Rochester Institute of Technology    [2]Queens University
[1]{kk3671, rxlics}@rit.edu    [2]mahdi.gilany@queensu.ca

## Abstract

Dropout regularization methods prune a neural network's pre-determined backbone structure to avoid overfitting. However, a deep model still tends to be poorly calibrated with high confidence on incorrect predictions. We propose a unified Bayesian model selection method to jointly infer the most plausible network depth warranted by data, and perform dropout regularization simultaneously. In particular, to infer network depth we define a beta process over the number of hidden layers which allows it to go to infinity. Layer-wise activation probabilities induced by the beta process modulate neuron activation via binary vectors of a conjugate Bernoulli process. Experiments across domains show that by adapting network depth and dropout regularization to data, our method achieves superior performance comparing to state-of-the-art methods with well-calibrated uncertainty estimates. In continual learning, our method enables neural networks to dynamically evolve their depths to accommodate incrementally available data beyond their initial structures, and alleviate catastrophic forgetting.

## 1   Introduction

Both dropout regularization and network depth are critical to the success of deep neural networks (DNNs) [1, 2, 3, 4]. Finely tuned or selected structures empower DNNs with proper model capacity that can not only efficiently capture statistical regularities in data but also avoid being deceived by random noise into "discovering" non-existent relationships.

Dropout as an effective regularization method sparsifies a neural network's structure by randomly deleting neurons along with their connections in training to prevent it from overfitting [5, 6]. Recent studies extend the method from a Bayesian perspective to allow automatic tuning of the dropout probability in large models [4, 7, 8, 9, 10, 11]. Although dropout and its variants can effectively govern model capacity, without uncertainty calibration deep models tend to be overly confident with their predictions [3, 12]. Probabilistic inference approaches for network structure selection propose to bypass or down-weight certain hidden layers to reduce training cost and build smaller models that perform just as well as larger ones [3, 13, 14, 15]. These methods lead to more efficient or better-calibrated models only by reducing the depth of a pre-determined network structure. They cannot scale the network structure up to accommodate incrementally available data beyond the upper limit of its capacity.

We thus propose a novel Bayesian model selection framework to jointly infer network depth and perform dropout regularization simultaneously. In particular, we model the depth of a network structure as a stochastic process by defining the beta process [16, 17] over the number of hidden layers to enable it to go to infinity in theory, as in Figure 1(a) (left). The beta process induces layer-wise activation probabilities, which allows its conjugate Bernoulli process to generate a binary

---

*Corresponding author

35th Conference on Neural Information Processing Systems (NeurIPS 2021).

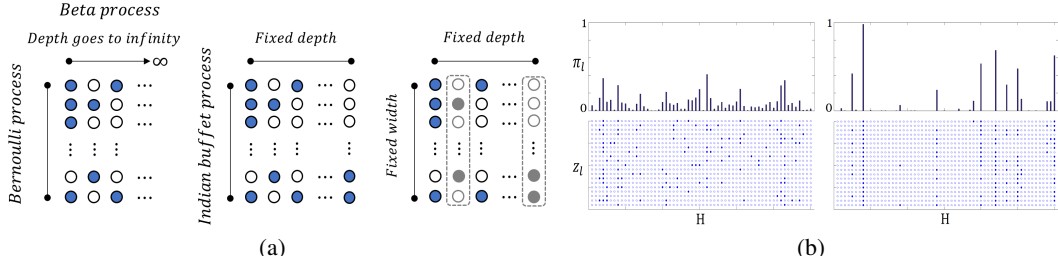

(a)                                                                                          (b)

Figure 1: (a) Demonstrations of our proposed framework (left), a dropout variant (middle), and typical structure selection methods (right). Our framework enables depth goes to infinity by modeling the number of hidden layers with a beta process. The dropout variant defines an Indian buffet process per layer to infer width, but depth is fixed [11, 20]. Structure selection methods can only reduce depth of a pre-determined network structure [2, 13, 14, 15]. (b) On top, random draws from two beta processes with $\alpha > \beta$ on the left and $\beta > \alpha$ on the right over hidden layer function space $\mathbf{H} = \{\mathbf{h}_l \,|\, l \to \infty\}$. An atom location is $\delta_{\mathbf{h}_l}$ indexing a hidden layer function $\mathbf{h}_l$, and the height denotes its activation probability $\pi_l$. For both cases, the conjugate Bernoulli processes at bottom are obtained by random filled dots $z_{ml} = 1$ with probability $\pi_l$ and empty dots $z_{ml} = 0$ with probability $1 - \pi_l$. So each column is a binary vector $\mathbf{z}_l$ corresponding to layer $l$.

vector per hidden layer to prune neurons for regularization, as in Figure 1(b). The probabilistic inference framework automatically balances network depth and dropout regularization by computing a marginal likelihood over the hidden layers and their neuron activations, and provides well-calibrated uncertainty estimates for predictions. The exact computation of the marginal likelihood is intractable due to the nonlinear nature of neural networks with a potentially infinite number of hidden layers. We thus employ the structured stochastic variational inference [18, 19] with a continuous relaxation on the Bernoulli variables to efficiently approximate the integral with a continuum of lower bounds. The relaxation allows to maintain differentiability and mitigate overfitting via model averaging of the sampled hidden layers and neuron activations. Benefiting from theoretical analysis, we readily integrate the joint inference with the parameter learning process through an alternating approach to efficiently update the parameters, and perform hidden-layer sampling and dropout regularization.

We analyze the behavior of our joint inference framework over multilayer perceptrons (MLPs) and convolutional neural networks (CNNs), and evaluate their performance across domains. The experiments show that our method leads to a compact neural network by balancing its depth and dropout regularization with uncertainty calibration, and achieves superior performance comparing to state-of-the-art dropout and structure selection methods. We also demonstrate our method on a continual learning task. By enabling both network depth and neuron activations to dynamically evolve to accommodate incrementally available data, we can alleviate catastrophic forgetting.

## 2 Related work

Dropout is an effective neural network regularization technique [6, 21]. By randomly pruning neurons and their connections from a network structure with a dropout rate during training, it can prevent DNNs from overfitting [22, 23, 24]. [7] interprets dropout from a Bayesian perspective to quantify uncertainty. Variational dropout proposes a stochastic gradient variational inference approach to learn the dropout rate from data with a constraint on large dropout rate values [8]. [9] extends the variational dropout to set unbounded individual dropout rate per weight. Concrete dropout proposes a continuous relaxation of the dropout's discrete masks to automatically tune the dropout probability, and obtain its uncertainty estimates [10]. Bayesian nonparametrics are also applied to extend the variational dropout approaches by defining an Indian buffet process (IBP) over neurons per hidden layer [11] or over channels in CNNs [20] to infer network widths, as in Figure 1(a) (middle). Another body of work leverage the non-parametric Bayesian framework to infer the dimensionality of latent representation encoded by DNNs [25, 26].

Structure selection methods improve training efficiency by reducing the depth of a pre-arranged network structure, as in Figure 1(a) (right). Some lead to well-performed smaller networks, and others achieve better uncertainty calibration. In order to build smaller models that perform just as

well, [2] proposes to reduce a large neural network by merging adjacent hidden layers with only linear relationships being activated in between. A stochastic depth method randomly drops a subset of hidden layers from a large network by bypassing them with an identity function in each mini-batch to speed up the training session, but it still deploys the large network structure at test time [13]. [15] shows that the multiplicative noise from dropout induces structured shrinkage priors over a neural network's weights, and they further extend the shrinkage framework based on ResNet structures [27] to model the probabilities of hidden layers being used. In particular, the work also indicates that the framework is equivalent to the stochastic depth regularization [13]. [14] proposes to tune a hidden layer's influence on a prediction by learning a bypass variable between adjacent-layer connections and skip connections with a variational Bayes method. The method also prunes neurons separately. [3] proposes a Bayesian model averaging method to down-weight deeper layers by directly connecting every hidden layer to the output layer, and achieve competitive performances with uncertainty calibration. [28] achieves model effectiveness and computational efficiency by inferring a distribution of connections and units in the context of local winner-takes-all DNNs with IBP.

## 3 The joint inference framework

We propose to model network depth as a stochastic process, and jointly perform dropout regularization upon the inferred hidden layers. Theoretical analysis shows that by approximating its marginal likelihood the inference framework achieves an optimal balance between network depth and neuron activations as Bayesian model selection.

### 3.1 Network structure with infinite hidden layers

Let $\mathbf{h}_l$ denote the $l$-th hidden layer composed of neurons (i.e., non-linear activation functions) $\sigma(\cdot)$. The neural network has the form:

$$\mathbf{h}_l = \sigma(\mathbf{W}_l \mathbf{h}_{l-1}) \bigotimes \mathbf{z}_l + \mathbf{h}_{l-1} \qquad l \in \{1, 2, ..., \infty\} \tag{1}$$

where $\mathbf{W}_l \in \mathrm{R}^{M \times M}$ is the layer $l$'s weight matrix with a Gaussian prior $p(\mathbf{W}) = \mathcal{N}(\mathbf{W}|0, s_w^2 \mathbf{I})$, $\bigotimes$ denotes element-wise multiplication of two vectors, and $M$ is the maximum number of neurons in a layer. For simplicity, we set $M$ to be the same for all hidden layers. In particular, we prune the $l$-th layer's outputs by multiplying them elementwisely by a binary vector $\mathbf{z}_l$ where its element variable $z_{ml} \in \{0, 1\}$. Each random variable $z_{ml}$ takes the value 1 with $\pi_l \in [0, 1]$ indicating activation probability of the $l$-th layer, as in Figure 1(b). The combination of the previous layer's outputs with the current layer's via skip connections not only avoids vanishing gradient but also propagates the output of the last hidden layer with activated neurons directly to the output layer $f(\cdot)$.[2]

Given a dataset $D = \{(\mathbf{x}_n, \mathbf{y}_n)\}_{n=1}^N$ with $N$ input-output pairs $(\mathbf{x}_n, \mathbf{y}_n)$ as training examples, for a regression task we express the likelihood of the neural network as:

$$p(D|\mathbf{Z}, \mathbf{W}) = \prod_{n=1}^N \mathcal{N}(\mathbf{y}_n | f(\mathbf{x}_n; \mathbf{Z}, \mathbf{W}), s^2 \mathbf{I}) \tag{2}$$

where $\mathbf{Z}$ is a binary matrix whose $l$-th column is $\mathbf{z}_l$, and $\mathbf{W} = \{\mathbf{W}_l\}$ denotes the set of the weight matrices. $s^2$ denotes likelihood noise with $\mathbf{I}$ as an identity matrix with the same dimensionality as the training examples' outputs. For classification tasks, we replace $f(\cdot)$ with a softmax function and the normal distribution with a multinoulli distribution.

### 3.2 Beta process over layer number

We treat the number of hidden layer functions $\mathbf{h}_l$ as a stochastic process by defining a beta process over their space $\mathbf{H} = \{\mathbf{h}_l\}_{l=1}^\infty$, as shown in Figure 1(b).

Conceptually, we generate a beta process $B$ from its base measure $B_0$ as $B \sim \mathrm{BP}(c, B_0)$ by drawing a set of samples $(\mathbf{h}_l, \pi_l) \in \mathbf{H} \times [0, 1]$ from a non-homogeneous Poisson process [16]. Let $B = \sum_l \pi_l \delta_{\mathbf{h}_l}$, where $\delta_{\mathbf{h}_l}$ is a unit point mass at $\mathbf{h}_l$. In our setting, a pair $(\mathbf{h}_l, \pi_l)$ corresponds to a hidden layer function $\mathbf{h}_l \in \mathbf{H}$ and its activation probability $\pi_l \in [0, 1]$. We define a conjugate

---

[2]A graphical demonstration of the network structure is in Appendix.

Bernoulli process $\mathbf{Z}_{m\cdot} \sim \text{BeP}(B)$ as $\mathbf{Z}_{m\cdot} = \sum_l z_{ml}\delta_{\mathbf{h}_l}$ at the same locations $\delta_{\mathbf{h}_l}$ as $B$ where $z_{ml}$ are independent Bernoulli variables with a probability that $z_{ml} = 1$ equals to $\pi_l$. As in (1), $z_{ml} = 1$ activates the $m$'th neuron in layer $l$.

Computationally, we employ the stick-breaking construction of beta-Bernoulli processes [29, 17] as

$$z_{ml} \sim \text{Ber}(\pi_l), \quad \pi_l = \prod_{j=1}^{l} \nu_j, \quad \nu_l \sim \text{Beta}(\alpha, \beta) \tag{3}$$

where $\nu_l$ are sequentially drawn from a beta distribution. According to the formulation, $\pi_l$ is decreasing with $l$. $\alpha$ and $\beta$ are the hyperparameters governing preference over a balance between network depth and dropout regularization. In particular, if $\alpha > \beta > 1$, the hidden layers tend to have lower activation probabilities with less number of activated neurons, as in Figure 1(b) left. The setting thus prefers a narrower but deeper network structure. On the other hand, $\beta > \alpha > 1$ favors shallower but wider network structure, where fewer hidden layers tend to have higher activation probabilities with more neurons being activated, as in Figure 1(b) right.

We thus define a prior over $\mathbf{Z}$ via the beta process as

$$p(\mathbf{Z}, \boldsymbol{\nu}|\alpha, \beta) = p(\boldsymbol{\nu}|\alpha, \beta)p(\mathbf{Z}|\boldsymbol{\nu}) = \prod_{l=1}^{\infty} \text{Beta}(\nu_l|\alpha, \beta) \prod_{m=1}^{M} \text{Ber}(z_{ml}|\pi_l) \tag{4}$$

### 3.3 Marginal likelihood over network structures

By combining the beta process prior over hidden layers in (4) with the neural network structure in (2), we can infer the number of hidden layers $L$ by integrating over $\mathbf{Z} = \{\mathbf{z}_l\}_{l=1}^{L}$ and $\boldsymbol{\nu} = \{\nu_l\}_{l=1}^{L}$:

$$p(D|\mathbf{W}, L, \alpha, \beta) = \int p(D|\mathbf{Z}, \mathbf{W})p(\mathbf{Z}, \boldsymbol{\nu}|\alpha, \beta)d\mathbf{Z}d\boldsymbol{\nu} \tag{5}$$

The exact computation of this marginal likelihood is intractable due to the non-linearity of the neural network and $L \to \infty$.

### 3.4 Efficient inference with SSVI

We employ structured stochastic variational inference (SSVI) to approximate the marginal likelihood with a variational lower bound (ELBO) retaining the dependence between network structures and hyperparameters [18, 19, 30]. In particular, we define the variational distribution as

$$q(\mathbf{Z}, \boldsymbol{\nu}|\{a_k\}_{k=1}^{K}, \{b_k\}_{k=1}^{K}) = q(\boldsymbol{\nu})q(\mathbf{Z}|\boldsymbol{\nu}) = \prod_{k=1}^{K} \text{Beta}(\nu_k|a_k, b_k) \prod_{m=1}^{M} \text{ConBer}(z_{mk}|\pi_k) \tag{6}$$

where $\pi_k = \prod_{j=1}^{k} \nu_j$, and $\{a_k, b_k\}_{k=1}^{K}$ are variational parameters of $q(\boldsymbol{\nu})$, and $K$ denotes a truncation level, which can be relaxed as in [31]. We relax the constraint of the discrete variables with continuous ones by reparameterizing the Bernoulli distribution into a concrete Bernoulli distribution [32, 33]:

$$\text{ConBer}(z_{mk}|\pi_k) = \tau \frac{\pi_k (z_{mk})^{-\tau-1}(1 - \pi_k)(1 - z_{mk})^{-\tau-1}}{(\pi_k(z_{mk})^{-\tau} + (1 - \pi_k)(1 - z_{mk})^{-\tau})^2} \tag{7}$$

where $\tau$ is a temperature controlling the distribution smoothness. We generate random samples from the distribution by first sampling from a logistic distribution as the external source of randomness and then putting the samples through a logistic function as follow:

$$z_{mk} = \frac{1}{1 + \exp(-\tau^{-1}(\log \pi_k - \log(1 - \pi_k) + \epsilon))} \qquad \epsilon \sim \text{Logistic}(0, 1) \tag{8}$$

This reparameterization of the Bernoulli distribution allows us to backpropagate the gradients of the lower bound with respect to the parameters while sampling from it.

We derive a lower bound of the log marginal likelihood in (5) as:

$$\log p(D|\mathbf{W}, L, \alpha, \beta) \geq \int q(\mathbf{Z}, \boldsymbol{\nu}) \log p(D|\mathbf{Z}, \mathbf{W})d\mathbf{Z}d\boldsymbol{\nu} + \int q(\mathbf{Z}, \boldsymbol{\nu}) \log \frac{p(\mathbf{Z}, \boldsymbol{\nu})}{q(\mathbf{Z}, \boldsymbol{\nu})}d\mathbf{Z}d\boldsymbol{\nu} \tag{9}$$

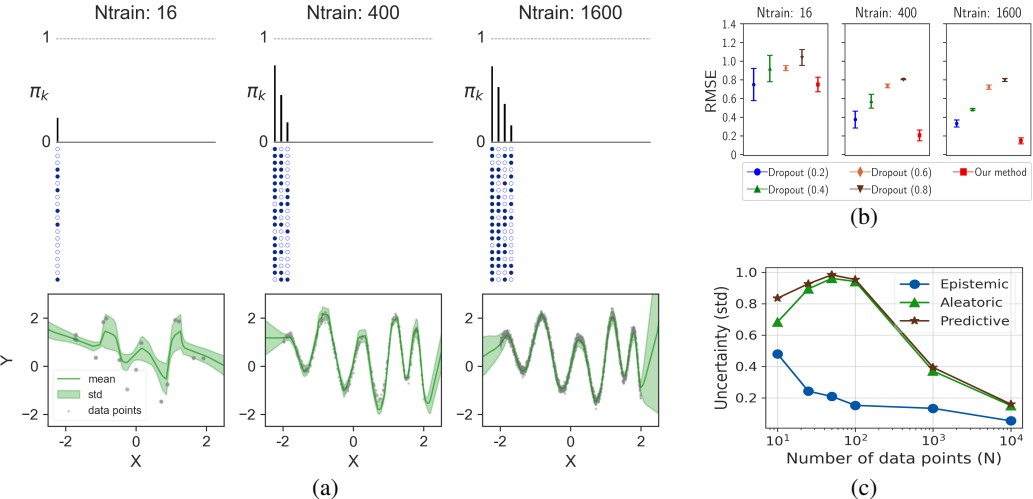

Figure 2: (a) Top row shows the activation probabilities $\pi_k$ (black bars) of the inferred hidden layers, and the neuron activations $\mathbf{Z}$ (filled dots denote activated neurons). The network becomes deeper with more activated neurons as the training dataset size increases. Bottom row shows the predictive distributions overlaying the data points, and the green bands are $\pm$one standard deviation over the predictions of 5 sampled network structures. (b) Predictive performance evaluation of our method and vanilla dropout with different dropout rates for the three cases. (c) Our method's estimates of different uncertainties as the number of data points increases.

The ELBO from the right-hand side of (9) becomes our objective function:

$$\mathcal{L}(\mathbf{Z}, \boldsymbol{\nu}, \mathbf{W}) = \mathbf{E}_{q(\mathbf{Z}, \boldsymbol{\nu})}[\log p(D|\mathbf{Z}, \mathbf{W})] - \mathrm{KL}[q(\mathbf{Z}|\boldsymbol{\nu})||p(\mathbf{Z}|\boldsymbol{\nu})] - \mathrm{KL}[q(\boldsymbol{\nu})||p(\boldsymbol{\nu})] \qquad (10)$$

The first term of the ELBO in (10) is the averaged log-likelihood. The second and the third terms are the KL divergence between the variational distribution and the true prior, respectively. In particular, for the second term we relax the prior $p(\mathbf{Z}|\boldsymbol{\nu})$ with a concrete Bernoulli as in (7) with different temperatures. For the third term the KL divergence between the variational distribution and the prior distribution over $\boldsymbol{\nu}$ is:

$$\begin{aligned}
\mathrm{KL}[q(\boldsymbol{\nu})||p(\boldsymbol{\nu})] = \sum_k &\ln\left(\frac{\mathrm{B}\left(\alpha, \beta\right)}{\mathrm{B}(a_k, b_k)}\right) + (a_k - \alpha)\,\psi(a_k) \\
&+ (b_k - \beta)\,\psi(b_k) + (\alpha - a_k + \beta - b_k)\,\psi(a_k + b_k)
\end{aligned} \qquad (11)$$

where $\psi(\cdot)$ denotes a di-gamma function.

**Theorem 1** *Assume* $\hat{\mathbf{W}}$ *is the optimized value of the variable, and the likelihood and the induced prior on* $\Pi_{\mathbf{Z}}$ *for* $\mathbf{Z}$ *are specified in (2) and (4), in the large sample limit, the limiting of the ELBO in (10) becomes equivalent to the Bayesian information criterion [34], such that* $BIC(\mathbf{Z}) = \log p(D|\mathbf{Z}, \hat{\mathbf{W}}) - \frac{|\mathbf{Z}|}{2}\log\frac{N}{2\pi}$, *where* $|\mathbf{Z}|$ *denotes the number of activated neurons in the network structure.*

The proof is in Appendix. This theorem indicates that optimizing the limiting case of our SSVI framework can therefore be seen as optimizing the popular model selection criteria. We adopt the implicit differentiation of beta distribution for backpropagation [35], which allows us to stochastically optimize $\mathcal{L}$ with respect to the variational parameters every time we sample a network structure. We iteratively maximize ELBO with respect to the variational parameters and weights $\mathbf{W}$, and approximate the ELBO using random samples from the variational distribution.

We approximate the posterior predictive distribution by sampling from the variational posterior distribution for predictions. Thus, the predictive distribution for new data $\mathbf{x}^*$ is

$$p(\mathbf{y}^*|\mathbf{x}^*, \hat{\mathbf{W}}, \{\hat{a}_k\}, \{\hat{b}_k\}) = \int p(\mathbf{y}^*|\mathbf{x}^*, \mathbf{Z}, \hat{\mathbf{W}})q(\mathbf{Z}, \boldsymbol{\nu}|\{\hat{a}_k\}, \{\hat{b}_k\})d\mathbf{Z}d\boldsymbol{\nu} \qquad (12)$$

where $\hat{\mathbf{W}}$ is the MAP estimate of the neural network weights based on mini-batch gradient descent.

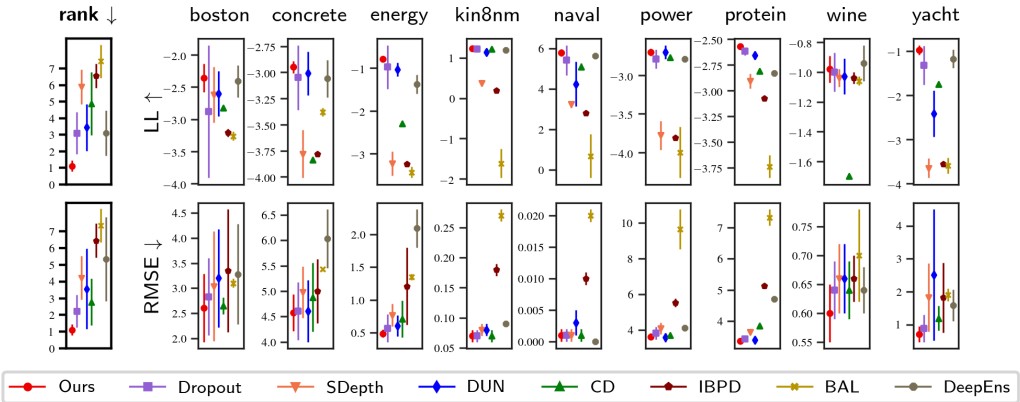

Figure 3: Mean values with $\pm$ one standard deviation for test log-likelihood (LL) and RMSE on UCI standard splits. Average ranks are computed across datasets. For LL, higher is better. For rank and RMSE, lower is better. The metrics are defined as in [3] with the codes generating the plots. Details are in Appendix.

## 4 Experiments

We analyze the behavior of our framework by applying it to MLPs and CNNs on a variety of tasks. We study how it enables network depth to adapt to dataset sizes, and capture different types of uncertainty on a synthetic dataset. Our framework outperforms state-of-the-art methods on the UCI datasets with uncertainty estimation. To obtain performance comparison, we perform Bayesian Optimisation and Hyperband (BOHB) to determine the best configurations for each method [36]. We also investigate the effect of truncation level $K$ and $M$, and show that their settings have no influence on the performance as long as they are reasonably large. The experiments with image datasets show that our framework improves CNN performances with more compact network structures and well-calibrated predictive distributions. Lastly, we demonstrate our method in two continual learning tasks, and show that by allowing network depth to dynamically augment to accommodate incrementally available information we can effectively alleviate catastrophic forgetting. [3]

### 4.1 Synthetic experiments

We analyze our method's behaviors given different dataset sizes, and its uncertainty estimation. We incrementally generate 20, 500 and 2000 data points from a periodic function [14]:

$$y = \sin(6x) + 0.4x^2 - 0.1x^3 - x\cos(9\sqrt{\exp(x)}) + \epsilon \qquad (13)$$

where $\epsilon \sim \mathcal{N}(0, 0.1^2)$ with 20% for validation. The data are shown in Figure 2(a) bottom row. We set the maximum number of neurons per layer $M = 20$, and use leaky ReLU activation functions in the input and hidden layers with batch normalization to retain stability. We simulate 5 samples per mini-batch to approximate the ELBO in (10), and evaluate convergence by computing the cross-correlations of the sample values over 3000 epochs.

As shown in Figure 2(a), to accommodate more information from larger datasets, the neural network becomes deeper with more activated neurons. Although we set the truncation level $K = 100$ for all the training cases, only the first few hidden layers have activated neurons (i.e., 1 layer, 3 layers, and 4 layers for the respective cases). Figure 2(a) bottom row shows how the predictive distributions in (12) are capturing the true function and estimating uncertainty as the dataset size increases. We compare our method's performance with vanilla dropout [6] whose backbone structure sizes are the same as the respective ones inferred by our method. The predictive performances on 700 held-out data points in Figure 2(b) indicate that by obtaining the critical balance between network depth and neuron activations our method achieves more stable performance with less variance on the small dataset, and significantly better performance on the larger ones.

---

[3]Implementation details of all the experiments are in Appendix.

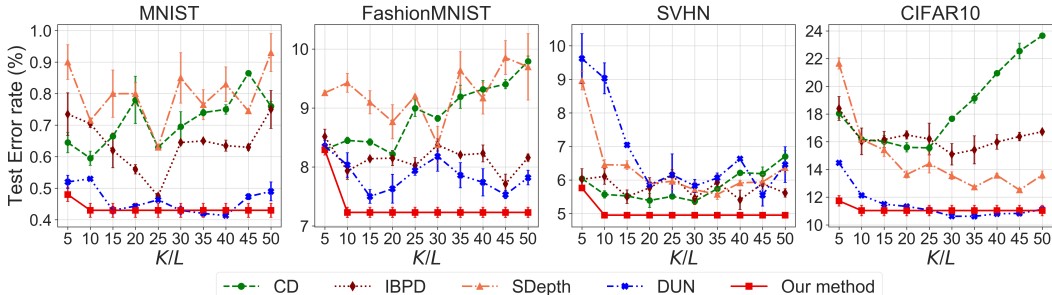

Figure 4: Analysis of the influence of our method's truncation level $K$ and other methods' backbone-structure depth $L$ in CNNs with the four image datasets. As long as $K$ is reasonably large ($\geq 10$), it no longer has an influence on our method's performances. On the contrary, the performance of the other methods depends on $L$. As $L$ becomes large, they tend to have an overfitting problem with increased test errors. By jointly inferring network depth and neuron activations, our method is robust to overfitting, and achieve the best performances.

The datasets generated with a known variance allow us to assess the different types of uncertainty of our method's predictions. In particular, the epistemic uncertainty is obtained by drawing multiple samples of the inferred depth and neuron activation, and evaluating them on the test set of 700 data points from the same data distribution. Figure 2(c) shows that the epistemic uncertainty decreases as the amount of data points increases. The aleatoric uncertainty approaching the true uncertainty (0.1) shows an increasingly improved estimate as more data is given. Our method's predictive uncertainty obtained by combining epistemic and aleatoric uncertainties converges to a constant value.

## 4.2 Performance comparisons on UCI datasets

We evaluate all methods on UCI datasets [37] using standard splits [38], and report their performance in terms of log-likelihood (LL) and root mean square error (RMSE). In particular, we compare the performances with vanilla dropout (Dropout), concrete dropout (CD) [10], Indian buffet process dropout (IBPD) [11], Bayesian architecture learning (BAL) [14], stochastic depth (SDepth) [13], depth uncertainty networks (DUNs) [3], and Deep Ensembles (DeepEns) [39]. We perform hyper-parameter optimization with Bayesian Optimisation and Hyperband (BOHB) to determine the best configurations including backbone-structure depths and hyperparameter settings for each method [36], as suggested in [3]. The detailed implementation is in Appendix.

In Figure 3, we rank the methods from 1 to 8 based on their mean performance across each dataset and metric, and report mean ranks with $\pm$ one standard deviation. Our joint inference framework outperforms other methods in terms of both test log-likelihood (LL) and RMSE by achieving the best rank across all datasets. In particular, LL measures both accuracy and uncertainty calibration [40]. The narrower standard deviations of our method indicate its robustness. The vanilla dropout achieves the second-best rank followed closely by the Deep Ensembles in terms of LL and concrete dropout in terms of RMSE.

## 4.3 Effect of truncation level

We further investigate the effect of truncation level $K$ on our method's performance, and compare it with the backbone-structure depth $L$ of dropout variants and structure selection methods.

All the methods have the same maximum width $M = 64$ (i.e., the maximum number of feature maps in convolutional layers). In Figure 4, we apply all the methods on CNNs with the backbone-structure depth $L$ and the truncation level $K$ over the range $L = K \in \{5, 10, 15, 20, 25, 30, 35, 40, 45, 50\}$ for classification tasks of four image datasets: MNIST [41], FashionMNIST [42], SVHN [43], and CIFAR10 [44]. When $L = K = 5$, all the methods have an underfitting problem due to the limited model capacity, although our method still achieves competitive or better performances. As the truncation level becomes reasonably large ($\geq 10$), it no longer affects our method's performance. The results are consistent with **Theorem 1**, as our joint inference essentially works as Bayesian model selection on a network structure truncation. On the contrary, the depths of backbone structure $L$

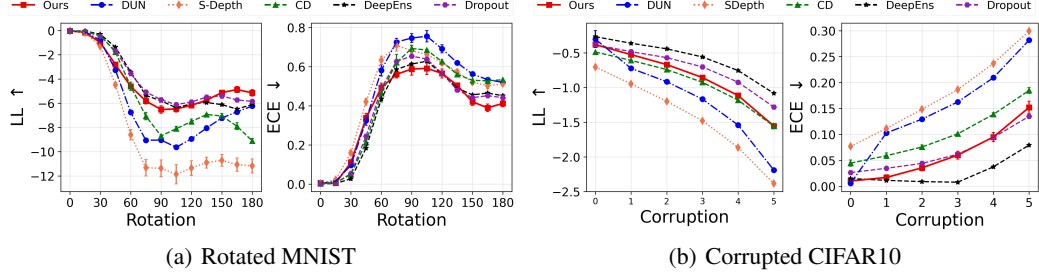

(a) Rotated MNIST          (b) Corrupted CIFAR10

Figure 5: Evaluation of uncertainty estimates for (a) MNIST for varying degrees of rotation and (b) CIFAR10 at varying degrees of corruption severities with log-likelihood (LL) and expected calibration error (ECE), as in [3].

affect the other methods to a great extent. The increase of the test errors shows that they all have an overfitting problem when the backbone-structure depths become large. Individual method reaches its own best performance, as $L$ increases. Our method outperforms them for all these cases with orders of magnitude less computing resources across the four datasets. The respective evaluations in Figure 6 (i.e., $M = 64$) show that our method only activates neurons in 6, 6, 10 and 6 hidden layers for predictions on MNIST, FashionMNIST, SVHN, and CIFAR10, respectively. Only $10\%$ of total neurons in the truncation level are activated for all the cases. This suggests by jointly inferring network depth and neuron activations our method is quite robust to overfitting, and the balance between the two also leads to superior performances with compact network structures. In contrast, the backbone-structure sizes of the dropout variants and structure selection methods have to be set carefully to determine the best configuration for a given dataset.

We also evaluate the uncertainty calibration of our framework and other methods with their best settings in Figure 4. In Figure 5(a), we train all methods on MNIST and evaluate their predictive distributions on increasingly rotated digits [45]. The uncertainty calibration of our method is more robust to the data with large angle rotations than other methods, and it achieves the performance with the lowest expected calibration error (ECE) [46]. In Figure 5(b), to make a fair and rigorous comparison as in [3], we also evaluate the corruption robustness of all methods on the CIFAR10 dataset with 16 types of algorithmically generated corruptions [47]. Each type of corruption has 5 levels of severity. The Deep Ensembles outperform other methods, and our model achieves comparable performance at all corruption levels.

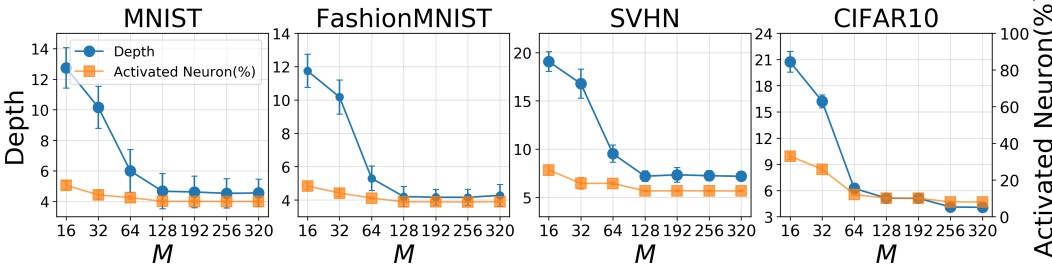

Figure 6: Influence of the maximum number of feature maps ($M$) on our method with four image datasets. When $M$ is small, we tend to have deeper network structures (blue). As $M$ becomes reasonably large (e.g., $\geq 128$), it tends not to have influence on the inference of network structure sizes. Meanwhile, the percentages of activated neurons in the truncation level are stable (orange).

### 4.4   Effect of $M$

We next assess the influence of the maximum number of neurons/feature maps ($M$) on our method with classification tasks on the four image datasets. Figure 6 shows the evolution of the number of inferred convolutional layers as $M$ increases. When $M = 16$, we tend to have deeper network structures to compensate for the relatively narrow layers. As $M$ increases, the structures become

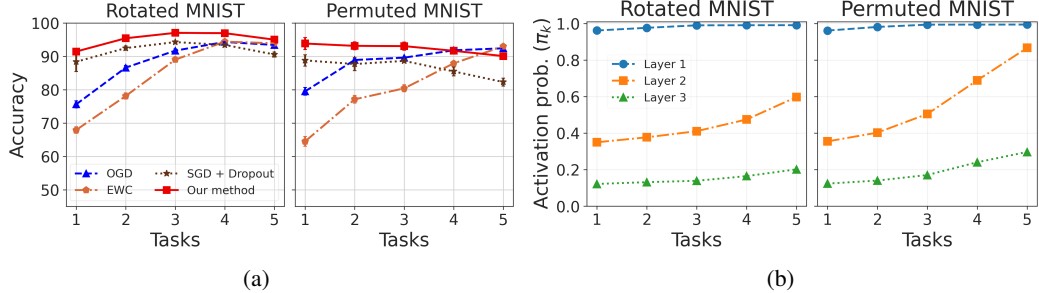

Figure 7: (a) The validation accuracy averaged over five runs for each task after training on all tasks in sequence. (b) Evolution of layer-activation probabilities ($\pi_k$) with the five tasks.

shallower. When $M$ is reasonably large, it has no influence on the inferred structure depth. The percentage of activated neurons in the truncation level remains relatively stable across different values of $M$. This suggests that our method can adapt both the network depth and the neuron activations to maintain the best performances as $M$ changes.

## 4.5 Continual learning

Continual learning is an important application since real-world tasks are dynamic and non-stationary. Machine learning models need to learn consecutive tasks without forgetting how to perform previously trained ones. Assuming the pre-determined backbone structures are sufficient to accommodate all information from the continual tasks, network regularization approaches alleviate catastrophic forgetting by regularizing the updates of neural weights [48]. Dropout and its variants are applied to learn an implicit gating mechanism that activates different gates for different tasks [49, 50]. However, the rigid backbone structure constrains the applicability of these methods in real-world settings.

We slightly modify our method to enable network depth and neuron activations to dynamically evolve to accommodate incrementally available data. Given a set of sequentially arriving datasets $\{D_t\}$ where each may contain a single datum, we update our ELBO in (10) as:

$$\mathcal{L}_t = \mathbf{E}_{q_t(\mathbf{z},\boldsymbol{\nu})}[\log p(D_t|\mathbf{Z},\boldsymbol{\nu},\mathbf{W})] - \mathrm{KL}[q_t(\mathbf{Z}|\boldsymbol{\nu})||q_{t-1}(\mathbf{Z}|\boldsymbol{\nu})] - \mathrm{KL}[q_t(\boldsymbol{\nu})||q_{t-1}(\boldsymbol{\nu})] \qquad (14)$$

by replacing the priors $p(\mathbf{Z}|\nu)$ and $p(\nu)$ with their variational approximation for the previous dataset $q_{t-1}(\mathbf{Z}|\nu)$ and $q_{t-1}(\nu)$. We initialize them as $q_0(\mathbf{Z}|\nu) = p(\mathbf{Z}|\nu)$ and $q_0(\nu) = p(\nu)$. The variational distribution in (6) after seeing the $t$-th dataset is recursively updated by taking the distribution after seeing the $(t-1)$-th dataset, multiplying by the likelihood and re-normalizing.

We compare the performance of our method with baseline continual learning methods including elastic weight consolidation (EWC) [48], orthogonal gradient descent (OGD) [51], and stochastic gradient descent with dropout (SGD + dropout). We conduct an extensive grid search on the hyperparameter setting of these methods, and evaluate them on two popular continual learning benchmarks: permuted MNIST [48] and rotated MNIST datasets [52]. Permuted MNIST at each time step $D_t$ consists of labeled MNIST images whose pixels undergo a fixed random permutation. Rotated MNIST is generated by the rotation of the original MNIST images. Figure 7(a) shows our method outperforms the other methods on the first four tasks, and is only slightly worse than some (i.e., SGD for Permuted MNIST). Over the five tasks, our method achieves the highest average accuracies ($95.16\%$ on Rotated MNIST and $92.34\%$ on permuted MNIST). Although some methods perform well on the fifth task after being trained on it, they fail to preserve the knowledge learned in previous tasks. In contrast, our method can preserve the knowledge from old tasks and achieves comparable or better accuracy on new tasks. Figure 7(b) shows the evolution of network structures for the five sequential tasks, as our method activates more neurons in the inferred hidden layers. In particular, the first hidden layer tends to be full from the first task with its activation probability close to one, the activation probabilities of the second and the third layers go up as the tasks are coming.

# 5   Conclusions

The proposed general joint inference framework can be applied to various neural networks. The experiments on MLPs and CNNs show that our method balances network depth and neuron activations to achieve superior performance. By enabling neural network structures to dynamically evolve to accommodate incrementally available data, we effectively alleviate catastrophic forgetting.

# 6   Funding

This material is based upon work supported by the National Science Foundation under Award No. 2045804 and Award No. 1850492.

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
