# Supplementary Material for Joint Inference for Neural Network Depth and Dropout Regularization

**Kishan K C**[1]     **Rui Li**[1]     **Mahdi Gilany**[2]
[1]Rochester Institute of Technology     [2]Queens University
[1]{kk3671, rxlics}@rit.edu     [2]mahdi.gilany@queensu.ca

## A   Proof of Theorem 1

To prove Theorem 1, let the binary matrix $\mathbf{Z}$ index the activated neurons with its non-zero elements, and we denote their weights as $\mathbf{W_Z}$. We employ a Gaussian approximation to the posterior distribution as follows. Let

$$p(\mathbf{W_Z}|D) = \frac{1}{p(D)}p(\mathbf{W_Z}, D) = \frac{1}{C}e^{-E(\mathbf{W_Z})} \tag{1}$$

where $E(\mathbf{W_Z})$ is equal to the negative log of the unnormalized log posterior:

$$E(\mathbf{W_Z}) = -\log p(\mathbf{W_Z}, D) \tag{2}$$

with $C = p(D)$ being the normalization constant. We expand $E(\mathbf{W_Z})$ around the mode $\hat{\mathbf{W}}_\mathbf{Z}$ with the Taylor series as

$$\begin{aligned}E(\mathbf{W_Z}) &\approx E(\hat{\mathbf{W}}_\mathbf{Z}) + (\mathbf{W_Z} - \hat{\mathbf{W}}_\mathbf{Z})^T g \\ &+ \frac{1}{2}(\mathbf{W_Z} - \hat{\mathbf{W}}_\mathbf{Z})^T H(\mathbf{W_Z} - \hat{\mathbf{W}}_\mathbf{Z})\end{aligned} \tag{3}$$

where $g$ is the gradient and $H$ is the Hessian of the energy function evaluated at the mode:

$$g = \nabla E(\mathbf{W_Z})|_{\hat{\mathbf{W}}_\mathbf{Z}} \qquad H = \frac{\partial^2 E(\mathbf{W_Z})}{\partial \mathbf{W_Z} \partial \mathbf{W}_\mathbf{Z}^T}|_{\hat{\mathbf{W}}_\mathbf{Z}} \tag{4}$$

The second term in (3) can be dropped, since the gradient term $(\mathbf{W_Z} - \hat{\mathbf{W}}_\mathbf{Z})^T g = 0$ due to the mode $\hat{\mathbf{W}}_\mathbf{Z}$. We thus have

$$\begin{aligned}p(\mathbf{W_Z}&|D) \\ &= \frac{1}{p(D)}p(\mathbf{W_Z}, D) \\ &= \frac{1}{C}e^{-E(\mathbf{W_Z})} \\ &\approx \frac{1}{C}e^{-E(\hat{\mathbf{W}}_\mathbf{Z})-0-\frac{1}{2}(\mathbf{W_Z}-\hat{\mathbf{W}}_\mathbf{Z})^T H(\mathbf{W_Z}-\hat{\mathbf{W}}_\mathbf{Z})} \\ &= \frac{e^{-E(\hat{\mathbf{W}}_\mathbf{Z})}}{C}\exp[-\frac{1}{2}(\mathbf{W_Z}-\hat{\mathbf{W}}_\mathbf{Z})^T H(\mathbf{W_Z}-\hat{\mathbf{W}}_\mathbf{Z})] \\ &= N(\mathbf{W_Z}|\hat{\mathbf{W}}_\mathbf{Z}, H^{-1}) \\ C &= p(D) \\ &= \int p(\mathbf{W_Z}, D)d\mathbf{W_Z} \\ &= e^{-E(\hat{\mathbf{W}}_\mathbf{Z})}(2\pi)^{\frac{|\mathbf{Z}|}{2}}|H|^{-\frac{1}{2}}\end{aligned} \tag{5}$$

35th Conference on Neural Information Processing Systems (NeurIPS 2021).

We abuse the notation a little by using $|\mathbf{Z}|$ to denote the number of non-zero elements.

The log marginal likelihood can thus be approximated with the Gaussian approximation as follows:

$$\log p(D) \approx -E(\hat{\mathbf{W}}_{\mathbf{Z}}) - \frac{1}{2}\log|H| + \frac{|\mathbf{Z}|}{2}\log 2\pi$$
$$= \log p(D|\hat{\mathbf{W}}_{\mathbf{Z}}) + \log p(\hat{\mathbf{W}}_{\mathbf{Z}}) - \frac{1}{2}\log|H| + \frac{|\mathbf{Z}|}{2}\log 2\pi \tag{6}$$

The penalization terms following $\log p(D|\hat{\mathbf{W}}_{\mathbf{Z}})$ are a measure of model complexity. The second term can be ignored by assuming a uniform prior. By approximating each $H_i$ by a fixed matrix $\hat{H}$ in the third term, $H = \sum_{i=1}^{N} H_i$, where $H_i = \nabla\nabla \log p(D_i|\mathbf{W}_{\mathbf{Z}})$, we have

$$\log|H| = \log|N\hat{H}|$$
$$= \log(N^{|\mathbf{Z}|}|\hat{H}|) \tag{7}$$
$$= |\mathbf{Z}|\log N + \log|\hat{H}|$$

where we assume $H$ is a full rank matrix. $\log|\hat{H}|$ term can be ignored due to its independence of $N$. We thus have the Bayesian information criteria (BIC) score:

$$\log p(D) \approx \log p(D|\hat{\mathbf{W}}_{\mathbf{Z}}) - \frac{|\mathbf{Z}|}{2}\log\frac{N}{2\pi} \tag{8}$$

Optimizing the ELBO produces the best approximation to the true posterior within the space of distributions, as well as the tightest lower bound on the true marginal likelihood. Our variational inference framework optimizes a lower bound to this BIC with respect to $\mathbf{Z}$ in order to have the most plausible network structure size.

## B  Algorithmic description

Our proposed method first draws $S$ samples of the network structures $\{\mathbf{Z}_s\}_{s=1}^{S}$ (network depth with layer-wise activation probabilities) from the variational distribution $q(\mathbf{Z}, \boldsymbol{\nu})$. The stick-breaking construction of beta-Bernoulli process induces that the probability of seeing activated neurons in hidden layers decreases exponentially with $K$, and we only need to retain the maximum number of layers with activated neurons up to the current iteration. With a reasonably large $K$ and a relatively small number of layers with active neurons, we obtain the number of hidden layers $l^c$ with activated neurons in the current iteration as:

$$l^c = \max_{l}\{l| \sum_{m=1}^{M} z_{ml} > \epsilon\} \tag{9}$$

where $\sum_{m=1}^{M} z_{ml}$ represents the total activation of neurons in layer $l$ based on samples, and $\epsilon$ is a threshold close to zero (0.01 for all experiments). We next compute the expectation of log-likelihood based on $S$ samples as:

$$\mathbb{E}_{q(\mathbf{Z},\boldsymbol{\nu})}[\log p(D|\mathbf{Z}, \mathbf{W})] = \frac{1}{S}\sum_{s=1}^{S}[\log p(D|\mathbf{Z}_s, \mathbf{W})] \tag{10}$$

Finally, we optimize the model parameters with respect to the ELBO using mini-batch stochastic gradient descent. We summarize our method as follows.

---

**Algorithm 1:** Training of our proposed method

---

**1 Input** $\{D_i\}_{i=1}^B$: $B$ mini batches of data
**2 Input** $S$: the number of samples of network structures $\mathbf{Z}$

  1: **for** i $= 1, \ldots, B$ **do**
  2:     Draw $S$ samples of network structures $\{\mathbf{Z}_s\}_{s=1}^S$ from $q(\mathbf{Z}, \boldsymbol{\nu})$
  3:     **for** s $= 1, \ldots, S$ **do**
  4:         Compute the number of layers $l^c$ from $\mathbf{Z}_s$ using (9)
  5:         Compute $\log p(D_i | \mathbf{Z}_s, \mathbf{W})$ with $l^c$ layers
  6:     **end for**
  7:     Compute $\mathbb{E}_{q(\mathbf{Z}, \boldsymbol{\nu})}[\log p(D_i | \mathbf{Z}, \mathbf{W})]$ using (10)
  8:     Compute ELBO $\mathcal{L}(\mathbf{Z}, \boldsymbol{\nu}, \mathbf{W})$
  9:     Update $\{a_k, b_k\}_{k=1}^{l^c}$ and $\{\mathbf{W}\}_{k=1}^{l^c}$ using backpropagation
10: **end for**

---

**Time complexity** For training a dropout neural network that has depth $L$ and width $M$, the time complexity is $O(NBLM^2)$ with $N$ training examples and $B$ epochs, given fixed batch size. Let $T$ be the time cost of a single NN forward pass, with $S$ samples our method is linearly scalable as $ST$.

## C   Experimental Setup

We define a neural network of the form

$$\mathbf{h}_l = \sigma(\mathbf{W}_l \mathbf{h}_{l-1}) \bigotimes \mathbf{z}_l + \mathbf{h}_{l-1} \qquad l \in \{1, 2, ..., L\} \tag{11}$$

for all experiments where $\sigma(\cdot)$ consists of multiple operations such as linear layer, convolutional layer, non-linear activation, Batch Normalization and so on (Figure 8). $L \to \infty$ for our method. We use same layer block for all methods for fair comparison.

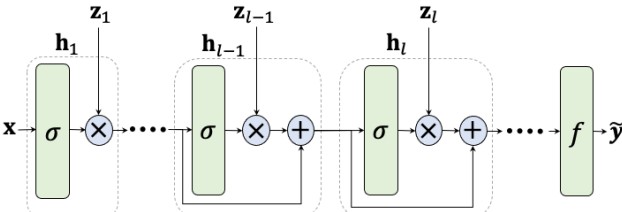

Figure 8: Block diagram of the network structure in (11). A hidden layer function $\mathbf{h}_l$ is pruned by multiplying their outputs by a binary vector $\mathbf{z}_l$ elementwisely. Since there are potentially infinite number of hidden layers between the input and the output layers, the skip connections feed the output of the last hidden layer with activated neurons to the output layer $f$. For CNNs, $\mathbf{z}_l$ regularizes feature maps in each block.

All of our method and other baselines: variants of dropout reguralization and structure selection methods are implemented in PyTorch [1] and all experiments are performed on a single NVIDIA GeForce RTX 2080Ti GPU. We next discuss the experimental settings for different experiments.

### C.1   Synthetic Experiments

We use fully connected neural networks with Leaky ReLU activation followed by Batch Normalization [2] and residual connections [3] for our method and vanilla dropout [4]. Formally, we define $\sigma(h_{l-1}) = \text{BatchNorm}(\text{LeakyReLU}(\text{Linear}(h_{l-1})))$ in (11). The maximum number of neurons (width) $M$ in hidden layer is 20 to observe how the number of inferred layers and their respective activation probabilities $\pi_k$ changes for tasks with different data sizes. The truncation level ($K$) for our method is set at 100 layers. The weight parameters are initialized with samples from the He

initialization [5]. The temperature parameter $\tau$ of Concrete Bernoulli distribution is set to $0.1$. We estimate the gradients of the likelihood term in ELBO via Monte Carlo estimation with 5 samples. Neural networks are optimized using Stochastic Gradient Descent (SGD) optimizer with momentum $0.9$, learning rate $3 \times 10^{-3}$, weight decay $10^{-5}$ and batch size of 200 for 3000 epochs. Furthermore, dropout probability was chosen from $p \in \{0.2, 0.4, 0.6, 0.8\}$. The depth ($L$) of network structure for dropout is set to the number of layers inferred by our method.

### C.2 Regression Experiments

#### C.2.1 Hyperparameter Optimization and Training

Previous works use Hyperparameter Optimization (HPO) techniques such as Grid Search, Bayesian Optimization (BO) [6], and Bayesian Optimization and Hyperband (BOHB) [7] to determine the best configurations for each method. For regression experiments, we follow DUN [8] for the setting of BOHB, and generate the optimized configurations for all methods for fair comparison. We employ the HpBandSter implementation of Bayesian Optimization and Hyperband (BOHB)[1]. We select the best configurations for each method on the standard splits [9] of UCI regression benchmark datasets [10] using the default settings, as shown in Table 1. Furthermore, we use the implementation of DUN[2] [8] to draw some of the plots.

Table 1: Bayesian Optimization and Hyperband settings

| Setting | Value |
|---|---|
| eta | 3 |
| min_points_in_model | None |
| top_n_percent | 15 |
| num_samples | 64 |
| random_fraction | 1/3 |
| bandwidth_factor | 3 |
| min_bandwidth | $1e-3$ |

For other hyperparameters of BOHB that vary across different datasets, we provide per dataset settings: min_budget and max_budget, validation proportion, and early stopping patience. Using the standard train-test splits provided by [9], we further split the original training data into a train-validation split and use the validation set to optimize hyperparameters. In particular, the last $N$ elements of the training set are considered as a validation dataset, where $N$ is calculated from the validation proportion given in Table 2. The training and validation datasets are normalized by subtracting the mean and dividing by the variance of the new training set. BOHB performs minimization on the negative test log-likelihood. During optimization, we perform early stopping with the patience values as shown in Table 2.

Table 2: Per-dataset hyperparameter optimization configurations

| Dataset | Min budget | Max budget | Early stop patience | Validation proportion |
|---|---|---|---|---|
| Boston | 200 | 2000 | 200 | 0.15 |
| Concrete | 200 | 2000 | 200 | 0.15 |
| Energy | 200 | 2000 | 200 | 0.15 |
| Kin8nm | 50 | 500 | 50 | 0.15 |
| Naval | 50 | 500 | 50 | 0.15 |
| Power | 50 | 500 | 50 | 0.15 |
| Protein | 50 | 500 | 50 | 0.15 |
| Wine | 100 | 1000 | 100 | 0.15 |
| Yacht | 200 | 2000 | 200 | 0.15 |

---

[1] https://github.com/automl/HpBandSter
[2] https://github.com/cambridge-mlg/DUN

Table 3: Hyperparameters optimized for each method

| Hyperparameters | Ours | Dropout | CD | IBPD | BAL | SDepth | DUN |
|---|---|---|---|---|---|---|---|
| Learning rate | ✓ | ✓ | ✓ | ✓ | ✓ | ✓ | ✓ |
| SGD Momentum | ✓ | ✓ | ✓ | ✓ | ✓ | ✓ | ✓ |
| Weight decay | ✓ | ✓ | ✓ | ✓ | ✓ | ✓ | ✓ |
| Number of Layers ($L$) | | ✓ | ✓ | ✓ | ✓ | ✓ | ✓ |
| Dropout probability ($p$) | | ✓ | | | | | |
| Length-scale ($l$) | | | ✓ | | | | |
| Concentration ($\alpha$) | | | | ✓ | | | |
| Layer skip probability ($\gamma^l$) | | | | | | ✓ | |

Table 4: BOHB Hyperparameter optimization configurations. All hyperparameters were sampled from uniform distributions.

| Hyperparameter | Lower | Upper | Default | Log | Data type |
|---|---|---|---|---|---|
| Learning rate | $1 \times 10^{-4}$ | 1 | 0.01 | True | float |
| SGD Momentum | 0 | 0.99 | 0.5 | False | float |
| Weight decay | $1 \times 10^{-6}$ | 0.1 | $5 \times 10^{-4}$ | True | float |
| Number of Layers ($L$) | 1 | 40 | 5 | False | int |
| Dropout Probability ($p$) | $5 \times 10^{-3}$ | 0.5 | 0.2 | True | float |
| Length-scale ($l$) | $1 \times 10^{-4}$ | 2 | $1 \times 10^{-2}$ | True | float |
| Concentration ($\alpha$) | 0.54 | 5 | 1.1 | True | float |
| Layer skip probability ($\gamma^l$) | $5 \times 10^{-3}$ | 0.99 | 0.1 | True | float |

Each method has a different set of hyperparameters to optimize (Table 3) and the configurations of these parameters to optimize with BOHB are provided in Table 4. Specifically, our proposed method infers the appropriate depth for a backbone structure that can go to infinity in theory. For a fair comparison with other variants of dropout and structure selection methods, we choose the depth at which each method performs best. To this aim, we optimize the depth for other methods.

All neural network models for regression experiments consist of a fully connected layer with $M = 100$ neurons followed by Leaky ReLU activation and Batch Normalization. We employ residual connections to allow all methods to better take advantage of depth. All the methods are trained using SGD with momentum and batch size of 128. We optimize learning rate, SGD momentum, and weight decay for all methods using BOHB.

For vanilla dropout experiments, we add dropout to each layer and the dropout probabilities have been optimized. Moreover, for the experimental setup of concrete dropout (CD) [11], we optimize the prior length scale $l$ that controls the regularization of the weights. Scaling the inputs by $\frac{1}{l}$ with a Gaussian prior $\mathcal{N}(0, \mathbf{I})$ over weights is equivalent to placing a prior $\mathcal{N}(0, \frac{\mathbf{I}}{l^2})$ over weights $\mathbf{W}$ instead [12]. Furthermore, a long length-scale $l$ results in the weights with low magnitude and thus lead to slow-varying induced functions. In contrast, a short length scale provides weights with high magnitude and results in functions with high frequencies. For dropout methods, 10 samples are used to compute the Monte Carlo estimate of test log-likelihood.

For Indian buffet process dropout (IBPD) [13] experiments, the concentration hyperparameter $\alpha$ controls the overall sparsity level of $\mathbf{Z}$ and $M$ indicates the number of neurons/filters to be pruned. We thus optimize the ratio $\alpha/M$ (denoted by $\alpha$ in Table 4 for simplicity) to obtain sparser models. The temperature parameter of concrete distribution is set to $\tau = 0.1$.

For Bayesian architecture learning (BAL) [14] experiments, we optimize skip probabilities ($\gamma^l$) for independent layer-wise skip connections to adapt the depth of the network according to data. However, BAL still relies on the depth of predefined backbone structure and appropriate choice of the depth plays a crucial role in the performance of BAL.

For stochastic depth experiments [15], we follow the settings suggested in [15] to define a simple decay rule from $p_0 = 1$ for the input layer to $p_L = 0.5$ for the last Residual block:

$$p_l = 1 - \frac{l}{L}(1 - p_L) \qquad (12)$$

This allows the model to bypass the layer $\mathbf{h}_l$ with probability $(1 - p_l)$, leading to the network of reduced depth and thus significantly speeds up the training process. During testing, the model uses the network structure with all layers to make predictions.

For DUN's experiments [8], a surrogate categorical distribution is defined over depth to explain the data with multiple depths by marginalization. We define uniform priors, assigning the same mass to each depth.

For deep ensembles, we train 5 independent neural networks and aggregate their predictions. The depth for each neural network in the ensemble is selected over the range $\{5, 10, 15, 20\}$.

### C.2.2   Evaluation Strategy

BOHB selects the best configuration for each method, dataset, and split. In addition, BOHB provides the number of epochs at which the optimal configuration was found. Using the best configurations of hyperparameters and epoch selected by BOHB using train-validation split, we train and evaluate the performance of our method and other baselines on standard $90\% - 10\%$ train-test splits [9] of UCI regression benchmark datasets [10]. Results are reported by training on each training split for each dataset and reporting the average metrics on the respective test split. We train each method on the training set (normalized to zero mean and unit variance), and the prediction is made by normalizing the test set based on train set statistics. The inverse transform is applied to the predictions and test metrics such as RMSE and log-likelihood are computed.

### C.3   Image Experiments

For image experiments, we consider four image datasets. Each dataset contains images associated with a label from 10 classes: MNIST with $60,000$ training and $10,000$ test examples of $28 \times 28$ grayscale images, FashionMNIST with $60,000$ training images and $10,000$ test examples of $28 \times 28$ grayscale image, SVHN with $73,257$ training and $26,032$ testing examples of $32 \times 32$ real-world RGB images, and CIFAR-10 with $50,000$ training and $10,000$ test examples of $32 \times 32$ RGB images. We normalize the datasets to have zero mean and unit variance and use $10\%$ of train data as validation.

All neural network models for image experiments consist of a convolutional layer with Leaky RELU activation followed by Batch Normalization [2] and residual connections [3] for all methods. In particular, for the input block, we use $5 \times 5$ kernel convolutional layer combined with Leaky ReLU activation functions and $2 \times 2$ max pooling. The output block is composed of a global averaging layer followed by a fully connected residual block and a linear layer. We apply the softmax operation to the output activation functions. Each hidden layer consists of a $3 \times 3$ convolutional layer followed by a Leaky ReLU activation and Batch Normalization. We use a fixed learning rate of $3 \times 10^{-3}$ with weight decay $10^{-6}$ and a batch size of 64. We evaluate our method with the width of the layer over the range $M \in \{64, 128, 192, 256, 320\}$. The experimental settings for other methods for image classification tasks are similar as in the previous section.

### C.4   Continual Learning

For all continual learning experiments, we use fully connected neural networks with $M = 100$ Leaky ReLU neurons and skip connections. For our method, we set hyper-parameters $\alpha = 1.1$ and $\beta = 1$ to encourage shallow neural network structures. For other methods, the number of layers is set to $L = 2$ layers based on the several continual learning benchmarks [16]. We also use learning rate decay with an initial learning rate of $0.1$ and exponential learning rate decay for dropout-based methods and our method to stabilize the training procedure [17].

We consider continual learning with 5 tasks on permuted MNIST dataset and rotated MNIST dataset. Each task of the permuted MNIST dataset is generated by shuffling the pixels of images with the permutation that is the same between images in the same task but is different across the tasks. Each task of the rotated MNIST dataset is generated by the continual rotation of the original MNIST images where each task applies image rotation (between 0 to 180 degrees) to the original images.

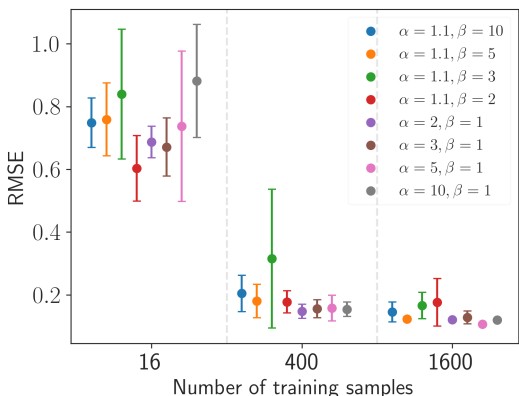

Figure 9: Predictive performance evaluation of our method with different hyperparameter settings for the three training cases.

Each model is trained on each task for five epochs to be consistent with several continual learning benchmarks [16]. We follow the experimental setup provided by [17] and define an SGD optimizer with a learning rate of $0.1$ and momentum of $0.8$. We further reduce the learning rate by $0.25$ after finishing each task. We optimize the dropout probability over the range $\{0.2, 0.3, 0.4, 0.5, 0.6\}$.

## D Calibrating Uncertainty

We consider regression on UCI regression benchmark datasets and classification on image datasets to evaluate uncertainty estimates. Two commonly used metrics are considered to evaluate uncertainty estimates with the formula as defined in [8]:

- **Test Log-Likelihood (LL)** (higher is better): Test log-likelihood measures both the accuracy of predictions and their uncertainty. We use log-likelihood for both regression and classification experiments.
- **Expected Calibration Error (ECE)** (lower is better): Expected Calibration Error [18, 19] is one of the popular metrics for uncertainty calibration and approximates the difference between predictive confidence and empirical accuracy. It is computed by partitioning predicted confidence $\hat{p}_i$ into $I$ equally-spaced bins and taking a weighted average of miscalibration in each bin:

$$\text{ECE} = \sum_{i=1}^{I} \frac{|B_i|}{N} |\text{acc}(B_i) - \text{conf}(B_i)| \tag{13}$$

with the number of samples $N$, accuracy of the bin $B_i$

$$\text{acc}(B_i) = \frac{1}{|B_i|} \sum_{i \in B_i} \mathbb{1}[y_i = \hat{y}_i]$$

and confidence of the bin $B_i$

$$\text{conf}(B_i) = \frac{1}{|B_i|} \sum_{i \in B_i} \hat{p}_i$$

$\hat{y}_i$ and $\hat{p}_i$ are the predicted labels and predicted confidence for a sample $i$. The difference between $\text{acc}(B_i)$ and $\text{conf}(B_i)$ in (13) for a given bin $B_i$ represents the calibration gap. We use ECE in addition to log-likelihood to measure calibration for image experiments.

## E Additional Results

### E.1 Analysis of Hyperparameters

We investigate the influence of hyperparameter settings $(\alpha, \beta)$ on the performance of our method with the synthetic dataset (Section 4.1). As shown in Figure 9, we investigate multiple hyperparameter

settings that favor shallow and wide network structures or deep and narrow network structures. The KL term in the variational lower bound acts like a regularizer on the model and dominates the ELBO for a small data regime because there is not enough data compared to model size. In contrast, the expectation of likelihood dominates the ELBO for tasks with a large dataset. The result suggests that our method's performance is not sensitive to the hyperparameter settings in general across the three training cases.

## E.2 Synthetic spiral datasets

We generate a 2d data set from a 2-armed spiral function with varying degrees of rotations ($180°$, $360°$, and $720°$) while keeping the number of training data points fixed to 200 [8]. We further draw 1800 samples as test set. For inference, we set hyper-parameters $\alpha = 1.1$ and $\beta = 2$ with maximum number of neurons in each hidden layer as $M = 20$. Other experimental settings are the same as in Section C.1.

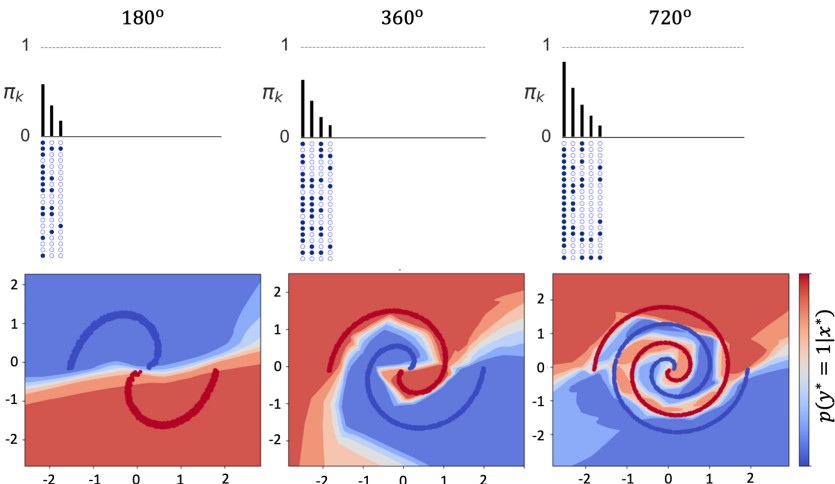

Figure 10: Top row shows the activation probabilities $\pi_k$ (black bars) of the inferred hidden layers, and the neuron activations $Z$ (filled dots denote activated neurons). The network becomes deeper with more activated neurons to adapt to the increasing complexity of the data generating process. The bottom row shows the predictive distributions overlaying the data points.

As shown in Figure 10, our model enables the neural network structure to grow by activating more layers and neurons to adapt to the increasing complexity of the underlying data generating process. In particular, it dynamically balances the depth and width of the network structure.

## E.3 Effect of M

We also evaluate the influence of the maximum number of neurons/feature maps ($M$) on the performance of our method with classification tasks on four image datasets (Section 4.4). From Figure 11, we observe that the performance of our model remains stable across different settings of $M$. This result thus suggests that our method adapts the depth of network structures and dropout regularization to maintain the balance leading to the best performances and is robust to the settings of $M$.

We investigate the evolution of activation probabilities ($\pi_k$) of sampled network structure as $M$ increases. Figure 12 shows the changes in activation level ($\pi_k$) over the training process as $M$ changes. During the training process, the activation of deeper layers decreases faster for larger values of $M$ and leads to compact network structures.

## E.4 Effect of truncation level

The performance of our model is not affected by a truncation level $K$ given that $K$ is reasonably large (Section 4.3). Furthermore, the truncation level also does not have any impact on the computational efficiency as shown in Figure 13. Since the activation probabilities $\pi_k$ decreases exponentially with $k$,

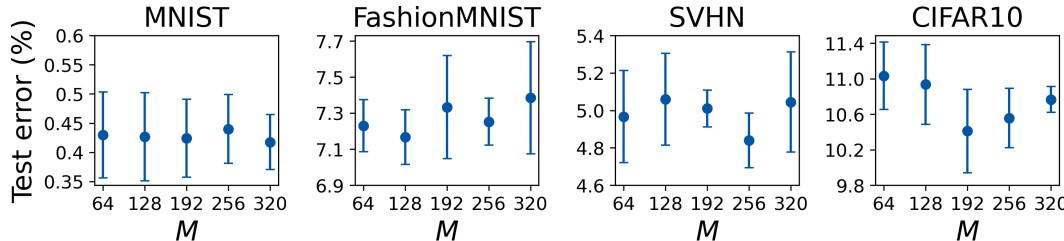

Figure 11: Influence of the maximum number of feature maps ($M$) on the performance of our method with four image datasets.

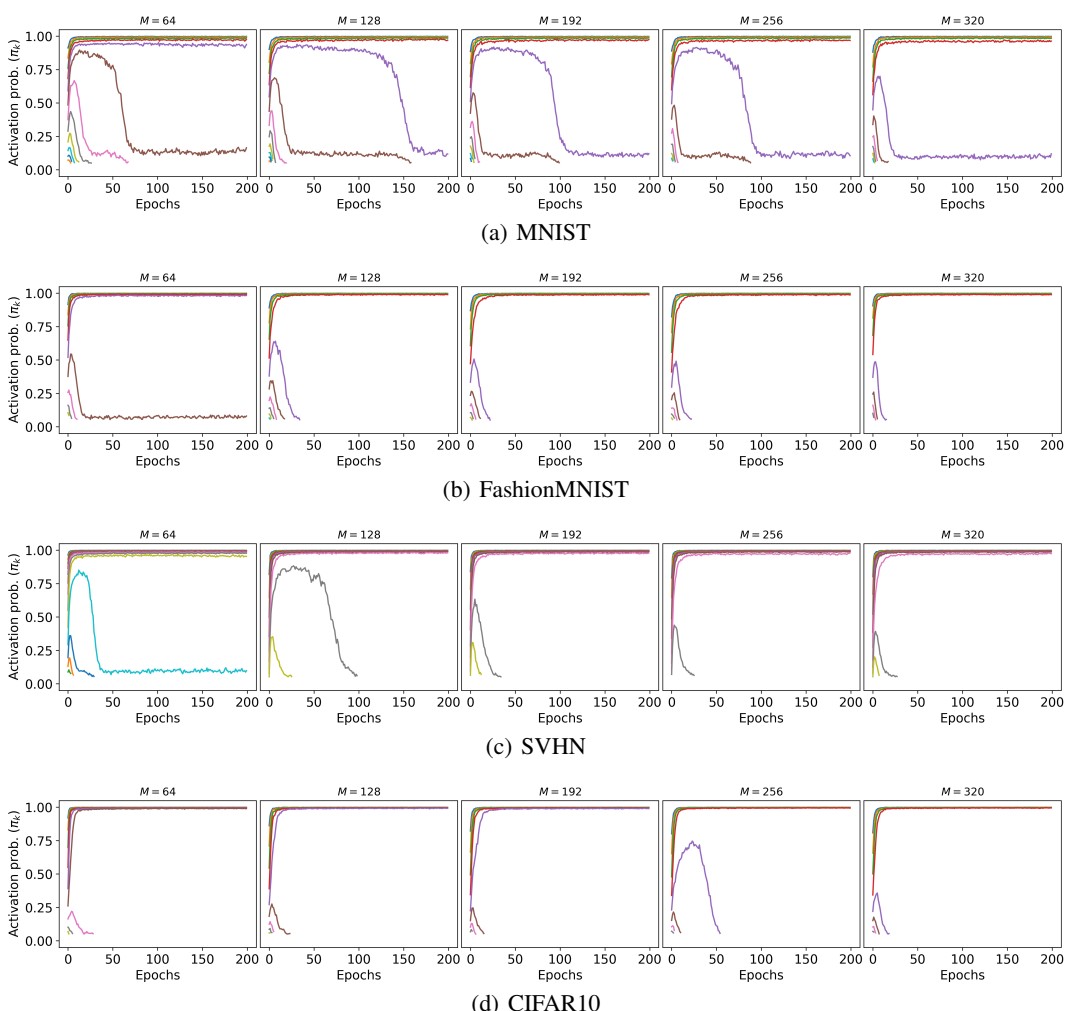

Figure 12: Evolution of activation probabilities as $M$ increases. Based on the stick-breaking construction of beta-Bernoulli process, the activation of the layer ($\pi_k$) is greater than or equal to the activation level of later layers ($\pi_{k+1}$). Top curve in each plot represents first layer and bottom layer represents last activated layer. The layers are dropped when the activation level $\pi_k < 0.05$.

with proper thresholding $\epsilon$, we only need to sample a relatively small number of hidden layers given a relatively larger truncation level $K$. From Figure 4, we observe that $K = 10$ is sufficiently large for MNIST. Figure 13 shows that truncation level $K \geq 10$ does not add any computational burden during both training and testing. We report the inference and prediction time for different models in Section E.5.

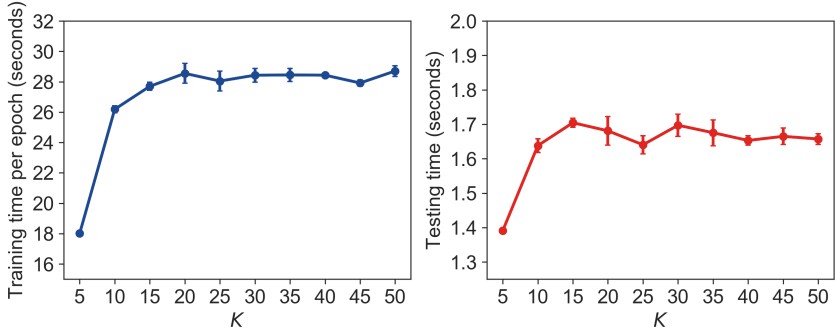

Figure 13: Training time per epoch and prediction time for model trained with different truncation level $K$ on MNIST dataset.

## E.5 Comparison of computational costs

For this experiment, we compare the inference and prediction time for different methods on MNIST and CIFAR10 dataset with the batch size 256. We measure the CPU times (in seconds) on a single NVIDIA RTX 2080Ti GPU. The inference and prediction times for all methods are reported in Table 5.

Table 5: The inference and prediction times of all methods. For inference, we report average time per epoch. The prediction time is what it takes to make prediction for all test samples.

| Methods | Inference | | Prediction | |
|---|---|---|---|---|
| | MNIST | CIFAR10 | MNIST | CIFAR10 |
| Our method | 18.314 | 27.232 | 1.016 | 1.016 |
| Dropout | 10.128 | 16.717 | 0.469 | 1.133 |
| DUN | 14.978 | 12.142 | 0.820 | 0.938 |
| Stochastic Depth | 22.538 | 41.125 | 1.406 | 1.953 |
| Concrete Dropout | 14.325 | 15.423 | 0.938 | 1.25 |
| IBP Dropout | 37.368 | 37.063 | 1.273 | 1.523 |
| Deep Ensembles | 50.642 | 83.583 | 5.01 | 8.281 |

Our method is no more than an order of magnitude faster than other dropout variants as shown in Table 5, since the stick-breaking construction of the beta process induces that $\pi_k$, the probability of seeing activated neurons in hidden layers, decreases exponentially with truncation level $K$. With proper thresholding, only a relatively small number of hidden layers are sampled in training. On the contrary, the adaptive sparsification method computes an Indian buffet process per hidden layer. This slows it down in training. For prediction, our method is about as fast as other dropout variants with small additional memory overhead. Deep ensembles require multiple forward passes which leads to slower prediction speed.

## E.6 Experiments on Out-of-Distribution data

For the problem of detecting out-of-distribution (OOD) samples, the dataset used in training is in-distribution dataset and the other datasets are considered as OOD dataset. For this experiment, we compute predictive entropy for the test dataset and consider a threshold-based classifier to classify the sample as in-distribution if the entropy is below some threshold. We then measure the area under the receiver operating characteristic curve (AUROC) and report in Table 6. First, we train our method on CIFAR10 (and MNIST) dataset and evaluate on in-distribution CIFAR10 (MNIST) test set and out-distribution SVHN (FashionMNIST) test set. The overall auroc of our method is competitive with other baselines including deep ensembles. Table 6 shows the performance of all methods based on the predictive entropy.

Second, we evaluate the methods on an OOD rejection task [8]. We sort the samples based on their entropy and reject different proportions of the samples with large entropy. We consider all predictions

Table 6: AUROC for in- vs out-distribution classification using predictive entropy

| Methods | MNIST vs FashionMNIST | CIFAR10 vs SVHN |
|---|---|---|
| Our method | $0.968 \pm 0.028$ | $0.869 \pm 0.014$ |
| DUN | $0.847 \pm 0.014$ | $0.838 \pm 0.033$ |
| Stochastic Depth | $0.912 \pm 0.001$ | $0.863 \pm 0.018$ |
| Concrete Dropout | $0.951 \pm 0.001$ | $0.855 \pm 0.009$ |
| Deep Ensembles | $0.991 \pm 0.002$ | $0.909 \pm 0.001$ |
| Dropout | $0.956 \pm 0.015$ | $0.899 \pm 0.004$ |

on OOD samples as incorrect and the rest of the samples are classified based on the model. For this experiment, we use CIFAR10 as in-distribution dataset and SVHN as out-of-distribution dataset. Deep Ensembles perform best in comparison to all other methods. Our method achieves similar performance for lower rejection rate and achieves superior performance for larger rejection rates when compared to DUN. The accuracy is reported in Figure 14.

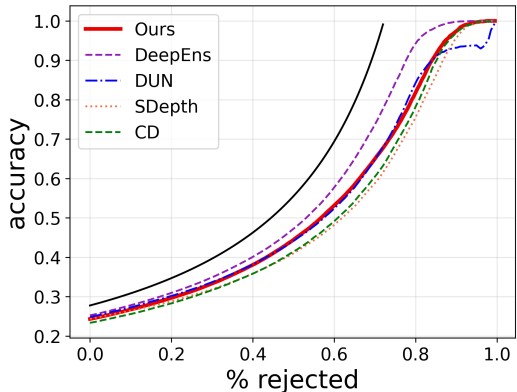

Figure 14: Rejection-classification plot for CIFAR10 (in-distribution) vs SVHN (out-distribution). The black solid line indicates theoretical maximum which indicates the accuracy when the model rejects all out-distribution samples and correctly classifies all in-distribution samples.

### E.7 Effect of Sample size $S$

Recent study [20] shows that by averaging an increasing number of samples for dropout regularization [4], the validation performance degrades and thus dropout loses the implicit regularization effect. In this experiment, we analyze the effect of sample size $S$ and investigate if our proposed model faces similar issue. We train our proposed model on two image datasets: CIFAR10 and SVHN with different values of $S$ over the range $\{1, 5, 10, 15, 20\}$ and visualize the validation loss over the epochs. Figure 15 shows that the validation errors decrease faster for larger values of $S$, but reach a similar level for all values of $S$. It indicates that the validation performance doesn't degrade as the sample size $S$ increases.

We further investigate the dynamics of $a_k$ and $b_k$ with respect to $S$ on synthetic periodic dataset. Within a reasonable range, the number of samples S does not have a significant impact on the performance of our method, as shown in Figure 16. As the reviewer suggested, we also investigate the dynamics of $a_k$ and $b_k$, the final performance of the model is comparable when $S \geq 5$. The influence of these hyper-parameter settings for the beta process prior is balanced out by the strong information from data. However, as S becomes larger, it incurs a higher computational cost.

### E.8 Feature maps visualization

Figure 17 demonstrates the learned feature maps of the last convolutional layers for variants of dropouts: vanilla dropout, concrete dropout, and Indian buffet process dropout on an image from the CIFAR10 dataset. With $M = 256$, our method only activates 29 feature maps compared to 256 for

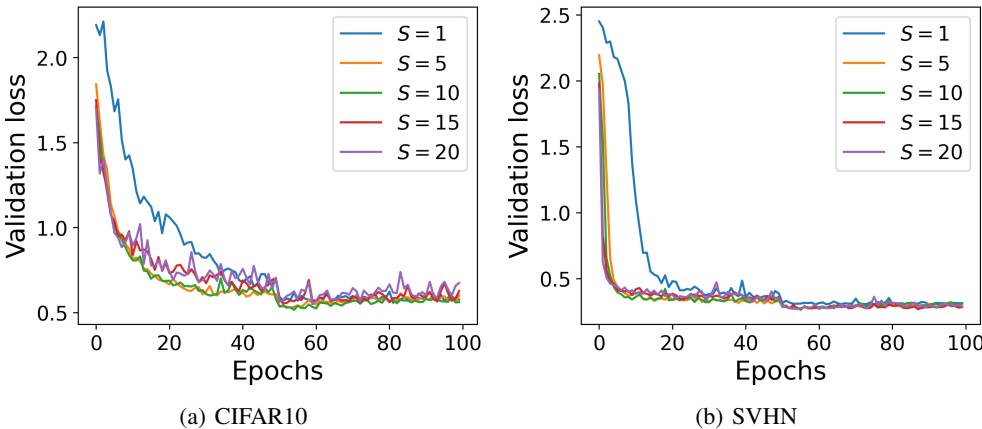

(a) CIFAR10          (b) SVHN

Figure 15: Validation loss over the epochs for our model trained with different sample size $S$ on (a) SVHN and (b) CIFAR10.

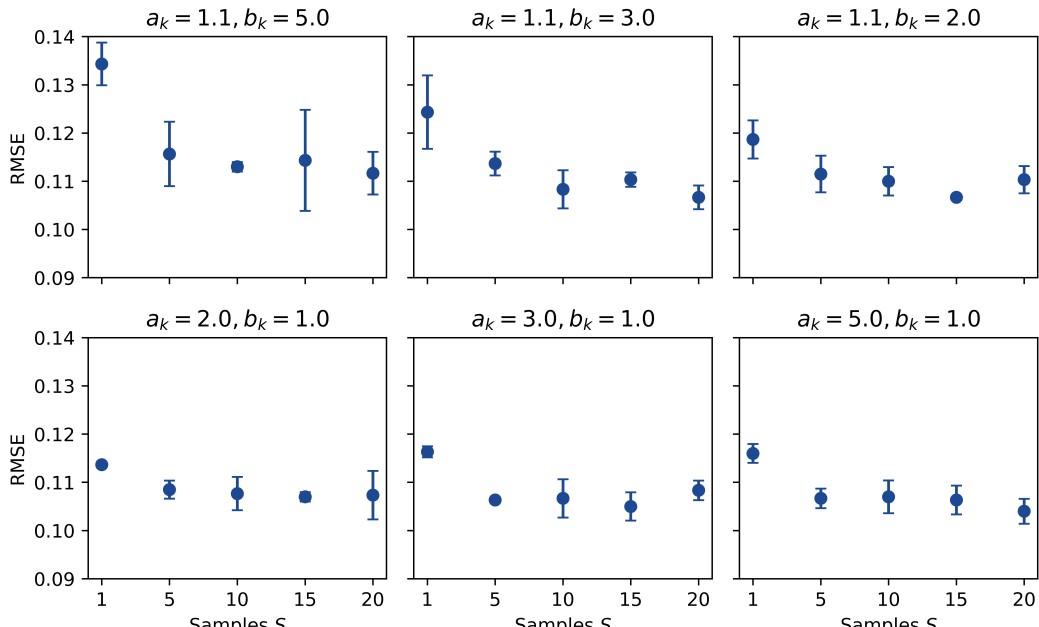

Figure 16: Dynamics of sample size $S$ with respect to $a_k$ and $b_k$ for synthetic experiment (Section 4.1). Within a reasonable range S does not have significant influence on the predictive performance.

dropout and concrete dropout, and $62$ for IBP dropout. Although vanilla dropout and concrete drop prune feature maps during training, they scale the weights and use all the feature maps for prediction. This indicates our method achieves superior performance with fewer but more informative feature maps.

### E.9 Regression Experiments

In Section 4.2, we compared the performance of our proposed model with Dropout, Concrete Dropout (CDropout), IBP Dropout, Stochastic Depth (SDepth), Bayesian Architecture Learning (BAL), Depth Uncertainty Networks (DUN) and Deep Ensembles (DeepEns) in terms of LL and RMSE. We report means and standard deviations of LL and RMSE along with their ranks based on the mean performance on UCI standard splits [9]. Table 7 and 8 show the values of LL and RMSE for Figure 3.

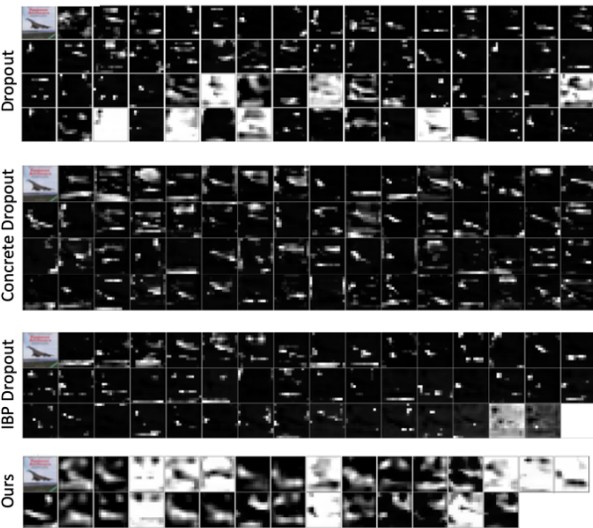

Figure 17: Demonstration of the subsets of feature maps from last hidden layer learned by vanilla dropout, concrete dropout, and Indian buffet process (IBP) dropout, and all the 29 feature maps from our method on CIFAR10. Our method has sparser but more informative feature maps in deeper layers.

Table 7: Mean values and standard deviations for test log-likelihood on UCI regression datasets across splits. Bold blue text denotes the best mean value for each dataset and each metric. Bold red text denotes the worst mean value. Note that in some cases the best/worst mean values are within error of other mean values.

| Method Dataset | Ours | Dropout | CDropout | IBPDropout | SDepth | BAL | DUN | DeepEns |
|---|---|---|---|---|---|---|---|---|
| Boston | $-2.36_{\pm 0.22}$ | $-2.88_{\pm 1.03}$ | $-2.82_{\pm 0.01}$ | $-3.21_{\pm 0.06}$ | $-2.62_{\pm 0.43}$ | $-3.26_{\pm 0.07}$ | $-2.60_{\pm 0.35}$ | $-2.41_{\pm 0.25}$ |
| Concrete | $-2.95_{\pm 0.06}$ | $-3.05_{\pm 0.31}$ | $-3.84_{\pm 0.01}$ | $-3.78_{\pm 0.01}$ | $-3.78_{\pm 0.23}$ | $-3.38_{\pm 0.04}$ | $-3.01_{\pm 0.21}$ | $-3.06_{\pm 0.18}$ |
| Energy | $-0.79_{\pm 0.07}$ | $-0.98_{\pm 0.51}$ | $-2.30_{\pm 0.01}$ | $-3.24_{\pm 0.01}$ | $-3.23_{\pm 0.28}$ | $-3.43_{\pm 0.13}$ | $-1.04_{\pm 0.16}$ | $-1.38_{\pm 0.22}$ |
| Kin8nm | $1.24_{\pm 0.02}$ | $1.23_{\pm 0.09}$ | $1.22_{\pm 0.04}$ | $0.19_{\pm 0.01}$ | $0.37_{\pm 0.07}$ | $-1.62_{\pm 0.36}$ | $1.15_{\pm 0.08}$ | $1.20_{\pm 0.02}$ |
| Naval | $5.79_{\pm 0.18}$ | $5.43_{\pm 0.74}$ | $5.10_{\pm 0.17}$ | $2.82_{\pm 0.02}$ | $3.24_{\pm 0.16}$ | $0.69_{\pm 1.08}$ | $4.25_{\pm 1.11}$ | $5.63_{\pm 0.05}$ |
| Power | $-2.70_{\pm 0.05}$ | $-2.79_{\pm 0.12}$ | $-2.77_{\pm 0.01}$ | $-3.81_{\pm 0.01}$ | $-3.78_{\pm 0.19}$ | $-4.00_{\pm 0.33}$ | $-2.70_{\pm 0.09}$ | $-2.79_{\pm 0.04}$ |
| Protein | $-2.57_{\pm 0.01}$ | $-2.62_{\pm 0.04}$ | $-2.81_{\pm 0.01}$ | $-3.08_{\pm 0.01}$ | $-2.91_{\pm 0.07}$ | $-3.74_{\pm 0.11}$ | $-2.66_{\pm 0.04}$ | $-2.83_{\pm 0.02}$ |
| Wine | $-0.98_{\pm 0.09}$ | $-1.00_{\pm 0.13}$ | $-1.70_{\pm 0.01}$ | $-1.04_{\pm 0.04}$ | $-1.04_{\pm 0.04}$ | $-1.06_{\pm 0.03}$ | $-1.03_{\pm 0.12}$ | $-0.94_{\pm 0.12}$ |
| Yacht | $-0.98_{\pm 0.11}$ | $-1.33_{\pm 0.44}$ | $-1.75_{\pm 0.01}$ | $-3.56_{\pm 0.02}$ | $-3.65_{\pm 0.22}$ | $-3.59_{\pm 0.18}$ | $-2.42_{\pm 0.52}$ | $-1.18_{\pm 0.21}$ |
| Rank | $1.11_{\pm 0.33}$ | $3.11_{\pm 1.27}$ | $4.89_{\pm 1.90}$ | $6.56_{\pm 0.73}$ | $5.89_{\pm 1.05}$ | $7.44_{\pm 1.01}$ | $3.44_{\pm 1.42}$ | $3.11_{\pm 1.36}$ |

Table 8: Mean values and standard deviations for test RMSE on UCI regression datasets across splits. Bold blue text denotes the best mean value for each dataset and each metric. Bold red text denotes the worst mean value. Note that in some cases the best/worst mean values are within error of other mean values.

| Method Dataset | Ours | Dropout | CDropout | IBPDropout | SDepth | BAL | DUN | DeepEns |
|---|---|---|---|---|---|---|---|---|
| Boston | $2.61_{\pm 0.68}$ | $2.83_{\pm 0.77}$ | $2.65_{\pm 0.17}$ | $3.35_{\pm 1.22}$ | $3.04_{\pm 1.09}$ | $3.10_{\pm 0.09}$ | $3.20_{\pm 0.98}$ | $3.28_{\pm 1.00}$ |
| Concrete | $4.58_{\pm 0.36}$ | $4.61_{\pm 0.57}$ | $4.88_{\pm 0.68}$ | $5.00_{\pm 0.63}$ | $4.98_{\pm 0.51}$ | $5.43_{\pm 0.04}$ | $4.61_{\pm 0.61}$ | $6.03_{\pm 0.58}$ |
| Energy | $0.49_{\pm 0.06}$ | $0.57_{\pm 0.21}$ | $0.71_{\pm 0.28}$ | $1.21_{\pm 0.59}$ | $0.77_{\pm 0.17}$ | $1.35_{\pm 0.05}$ | $0.61_{\pm 0.16}$ | $2.09_{\pm 0.29}$ |
| Kin8nm | $0.07_{\pm 0.01}$ | $0.07_{\pm 0.01}$ | $0.07_{\pm 0.01}$ | $0.18_{\pm 0.01}$ | $0.08_{\pm 0.01}$ | $0.27_{\pm 0.01}$ | $0.08_{\pm 0.01}$ | $0.09_{\pm 0.00}$ |
| Naval | $0.00_{\pm 0.00}$ | $0.00_{\pm 0.00}$ | $0.00_{\pm 0.00}$ | $0.01_{\pm 0.00}$ | $0.00_{\pm 0.00}$ | $0.02_{\pm 0.00}$ | $0.00_{\pm 0.00}$ | $0.00_{\pm 0.00}$ |
| Power | $3.61_{\pm 0.19}$ | $3.82_{\pm 0.35}$ | $3.70_{\pm 0.04}$ | $5.52_{\pm 0.24}$ | $4.08_{\pm 0.33}$ | $9.64_{\pm 1.11}$ | $3.57_{\pm 0.25}$ | $4.11_{\pm 0.17}$ |
| Protein | $3.37_{\pm 0.03}$ | $3.43_{\pm 0.07}$ | $3.85_{\pm 0.02}$ | $5.13_{\pm 0.03}$ | $3.66_{\pm 0.08}$ | $7.32_{\pm 0.26}$ | $3.40_{\pm 0.06}$ | $4.71_{\pm 0.06}$ |
| Wine | $0.60_{\pm 0.05}$ | $0.64_{\pm 0.05}$ | $0.64_{\pm 0.05}$ | $0.66_{\pm 0.04}$ | $0.66_{\pm 0.06}$ | $0.70_{\pm 0.08}$ | $0.66_{\pm 0.06}$ | $0.64_{\pm 0.04}$ |
| Yacht | $0.71_{\pm 0.24}$ | $0.88_{\pm 0.41}$ | $1.19_{\pm 0.38}$ | $1.82_{\pm 1.06}$ | $1.82_{\pm 1.04}$ | $1.90_{\pm 0.19}$ | $2.51_{\pm 1.99}$ | $1.58_{\pm 0.48}$ |
| Rank | $1.11_{\pm 0.33}$ | $2.22_{\pm 0.97}$ | $2.78_{\pm 1.39}$ | $6.44_{\pm 1.01}$ | $4.22_{\pm 1.30}$ | $7.33_{\pm 1.00}$ | $3.56_{\pm 2.40}$ | $5.33_{\pm 2.5}$ |