# OpenReview forum: "Joint Inference for Neural Network Depth and Dropout Regularization"
_NeurIPS.cc/2021/Conference — NeurIPS 2021 Poster_

### Official Review · Reviewer_FqX9 · 2021-07-11

**Rating:** 7
**Confidence:** 2

**Summary:**

The paper proposes a new method for training deep neural networks, where the parameters controlling model depth and activation 'drop' probabilities are being trained and updated, as well as the model's parameters. After developing a theoretical framework where the model structure is treated as a stochastic process, they run a series of experiments to evaluate it over a series of tasks, and present promising results regarding their model's ability to estimate uncertainty.

**Limitations And Societal Impact:**

1. [Request] Section 3.2: Please provide more early intuition for the usage of Beta Process.

2. [Request] The number of samples ($S=5$) used at Algorithm 1 seems arbitrary. Furthermore, as a key component in the method, I would be interested to see how it was chosen. Can you run an ablation study to show its effect? I would be interested to learn how the dynamics of $a_k, b_k$ change in accordance to $S$.

3. [Question] Please address the points I have made about the dropout scalar factor, batch-norm, and implicit regularization.

4. Please specify the limitations of the new method. What is the computational cost?

5. [Question] Please comment on the baselines selected for the UCI test. Are the baselines sufficient for the claim of surpassing
SOTA?

**Main Review:**

**Originality:**

The method presented in the paper is new. Several similar methods are discussed under 'prior work', and the differences are highlighted.

**Quality:**

The submission appears to be technically sound.

In section 3, some of the decisions made felt arbitrary and I would appreciate more care being given for their advantages and disadvantages. (See limitation: 1)

Several things struck me as odd with the method:

1. The dropout variant used in the paper doesn't include the 1/(1-p) factor which we know to be vital for standard dropout. Dropout is still used during inference so it may be less important, but it seems to me that the factor is still necessary for training since the value of the drop-rate is inconsistent.

2. The paper used batch-norm liberally for experiments (4), but its effect wasn't addressed in section 3 and I worry it could be substantial. Regularization effect aside, it can also affect the training's dynamics, especially if the dropout scalar factor is missing.

3. In Algorithm 1, $S$ samples of networks (Depth + Dropmask) are being drawn, and their results are averaged to estimate log-likelihood (for the ELBO). While it's not completely the same, it has similarities with $\text{Dropout}_K$, an experimental regularization that was explored in this paper: [2]. Paper [2], however, showed that $\text{Dropout}_K$ is suboptimal, since by averaging over different dropout masks you lost the *implicit* regularization of dropout. I worry that this may happen in this paper as well (this can explain why the chosen value for $S$ is so low). The good news is that even if this is a problem, this could possibly be fixed with the regularization term they propose.

The experimental section is well organized, and supported by complementary details in the supplementary. The authors claim superior performance when comparing their method to state-of-the-art methods albeit I was not convinced that the baselines were good enough for this assessment. (see: A Survey of Uncertainty in Deep Neural Networks[3], Table IV).

One more thing I felt was missing is an illustration of the dynamics of $a_k, b_k$ during training. As is, I am not confident enough that the motivations presented for designing the new method align with what it does in practice.

The submission included code, which allowed easy recreation of the synthetic experiment. I ran few sanity checks (Like detaching loss components and monitoring network depth)-- they all checked out, but they were somewhat limited by the easy benchmark.

**Clarity:**

The paper is well written.

Both the introduction and related work sections detail the relevant work from the machine learning aspect. The relevant background regarding the mathematical aspects (like [1]), however, is only briefly mentioned. This makes the later understanding of the introduced method harder for the readers who are not familiar with this works.

Algorithm 1 (in the Appendix) ("Training of our proposed network") was very helpful for understanding section 3 from a birds' eye view, consider incorporating it into the main text.

**Significance:**

The submission addresses difficult tasks (uncertainty estimation, continual learning), with a new approach. While I was not convinced of the practicality of the approach as is, others are likely to use/ build upon this work.

[1] https://arxiv.org/abs/1106.0539
[2] https://arxiv.org/abs/2002.12915
[3] https://arxiv.org/abs/2107.03342

**Time Spent Reviewing:**

12

---

> ### Author Response · Authors · 2021-08-10
> **We thank reviewer for your essential and constructive comments. Our detailed response is as below.**
>
> - **No scaling of dropout rate**   The scaling factor $\frac{p}{1-p}$ is necessary for dropout because it deploys the whole pre-determined network structure weighted by the scaling factor for prediction. Our method only uses a sampled structure with activated neurons, rather than the whole truncation, for prediction. So our network structure for prediction does not need such a scaling factor. We will clarify this in the camera-ready version.
>
> - **Effect of batch norm** We agree with the reviewer that batch norm may play a role in regularization. On the other hand, all the baseline methods employ Batch normalization in their algorithms. So, the comparison is still fair. To further study the influence, we re-run our algorithm with/without Batch Norm for the toy dataset. The final performance is comparable with the one using the Batch norm  (Table 7). We will clarify it in the camera-ready version.
>
>     **Table 7**: Test RMSE on the synthetic dataset (Section 4.1) with and without Batch Normalization
>
>     | With Batch Norm | Without Batch Norm |
>     | :-------------: | :----------------: |
>     | $0.107\pm0.002$ | $0.108 \pm 0.003$​  |
>
>
>
> - **Implicit regularization of dropout** Dropout$_{K}$ study shows that by averaging an increasing number of dropout samples, the validation performance degrades, and thus dropout loses the implicit regularization effect [Wei et al. 2020]. We conduct additional experiments to investigate the influence of our sample size. [Figure 4](https://www.dropbox.com/s/3g8yv0h7f43gkgz/figure_4.pdf?dl=0) shows that the validation performance of our proposed method does not degrade as $S$ increases. This is because the samples in our algorithms are to approximate the posterior of structure sizes for inference. However, we only use one sample for prediction. We will clarify it in the camera-ready version.
>
> - **Choice of S**  Within a reasonable range, the number of samples S does not have a significant impact on the performance and computational cost of our method, as shown in [Figure 5](https://www.dropbox.com/s/vrpgfjvhp6n9dzg/figure_5.pdf?dl=0). As the reviewer suggested, we also investigate the dynamics of $a_k$ and $b_k$, the final performance of the model is comparable when $S \geq 5$. The influence of these hyper-parameter settings for the beta process prior is balanced out by the strong information from data. However, as S becomes larger, it incurs a higher computational cost.
>
> - **Intuition of beta process** The main motivation for us to adopt the *beta process* to model the number of hidden layers is because it is a complete random measure. It assumes the activation probabilities of hidden layers are independent, each of which is in [0,1]. For comparison, The *Dirichlet process* has a constraint that the activation probabilities of all the hidden layers have to sum to 1. It implies that if there are some hidden layers with higher activations, the others have to be less activated, which is counterintuitive. The *Beta process* is derived from a *Poisson process* which is another complete random measure, and they share similar characteristics. *Indian buffet process* is a marginalized version of *beta-Bernoulli processes* over probabilities, and it usually models a fixed number of objects with infinite features. With an efficient inference algorithm, we consider the *beta process* as a sensible choice. For our future research, we will also investigate some recent stochastic processes (e.g., random function priors). We will clarify it in the camera-ready version.
>
> - **Computation** some complexity analysis is in the Appendix Line 36-38. For training a dropout neural network that has depth $L$​ and width $M$​, The time complexity is $O(NBLM^2)$​ with $N$​ training examples and $B$​ epochs, given a fixed batch size. Let $T$​ be the time cost of a single NN forward pass, with $S$​ samples our method is linearly scalable as $ST$​. Our method is no more than an order of magnitude faster than other dropout variants as shown in Table 1. Since the stick-breaking construction of beta process induces that $\pi_k$​, the probability of seeing activated neurons in hidden layers, decreases exponentially with the truncation level $K$​. With proper thresholding, only a relatively small number of hidden layers are sampled in training. On the contrary, the adaptive sparsification method computes an Indian buffet process per hidden layer. This slows it down in training. For prediction, our method is about as fast as other dropout variants with a small additional memory overhead. Deep ensembles require multiple forward passes which leads to slower prediction speed. We'll clarify these results in the camera-ready version, as the reviewer suggested.
>
>      **Table 1:** The inference and prediction times of all methods with batch size 256. We measure the CPU times (in seconds) for MNIST and CIFAR-10 data sets on a single NVIDIA RTX 2080Ti GPU. For deep ensembles, we evaluate the neural network with depth $\{5, 10, 15, 20\}$, and 5 neural networks are averaged for each case to determine the best configuration. For inference, we report the average time per epoch. The prediction time is what it takes to make predictions for all test samples.
>
>     |     Methods      | Inference |             | Prediction |             |
>     | :--------------: | :-------: | :---------: | :--------: | :---------: |
>     |                  | **MNIST** | **CIFAR10** | **MNIST**  | **CIFAR10** |
>     |    Our method    | $18.314$  |  $27.232$   |  $1.016$   |   $1.016$   |
>     |     Dropout      | $10.128$  |  $16.717$   |  $0.469$   |   $1.133$   |
>     |       DUN        | $14.978$  |  $12.142$   |  $0.820$   |   $0.938$   |
>     |  Deep Ensembles  | $50.642$  |  $83.583$   |   $5.01$   |   $8.281$   |
>     | Stochastic Depth | $22.538$  |  $41.125$   |  $1.406$   |   $1.953$   |
>     | Concrete Dropout | $14.325$  |  $15.423$   |  $ 0.938$  |   $1.25$    |
>     |   IBP Dropout    | $37.368$  |  $37.063$   |  $1.273$   |   $1.523$   |
>
> - **Baselines for UCI data sets** The competing methods we evaluated on UCI are the state-of-the-art dropout variants and structure selection methods. The results shows that our method achieves significant better overall performance.
>
> - **Clarity** As suggested by the reviewer, we will detail the background review, and incorporate Algorithm 1 into the main text.

---

> > ### Comment · Reviewer_FqX9 · 2021-08-17
> > **Thank you for the response**
> >
> > **Dropout factor**: I agree that the original use of the dropout factor $\frac{p}{1-p}$ is not directly relevant for this work. However, with the value of the dropout changing from iteration to iteration, I can still see its potential use to keep the variances of the network in check.
> >
> > **Regarding Implicit regularization**: That's a good indication, but I would still like this issue to be investigated further. The synthetic implicit regularization suggested by the paper depends on the Loss Jacobean, and while I don't have a good estimation of the implications of this, I suspect it might be weaker for the case of mnist. I think that accounting for this has the potential of strengthening the paper.

---

> > > ### Author Response · Authors · 2021-08-25
> > > **We thank the reviewer for your further comments**
> > >
> > > - **Dropout factor**: We will investigate the potential use of the dropout scale factor in our method as suggested by the reviewer.
> > >
> > > - **Regarding Implicit regularization**: As suggested by the reviewer, we further conducted the experiments to investigate the influence of sample size $S$ on two image datasets: SVHN and CIFAR10. [Figure 7](https://www.dropbox.com/s/8kw7j7kgylygzn7/figure_7.pdf?dl=0) shows that the validation performance of our proposed method does not degrade as $S$ increases. In the camera-ready version, we will discuss the work and add the new results into the appendix. In the future, we will do further investigation to see if we can incorporate the idea to strengthen our method.

---

> > > > ### Comment · Reviewer_FqX9 · 2021-08-26
> > > > **Implicit regularization**
> > > >
> > > > Thanks.
> > > > It does appear as if S=5 is somewhat superior to S=10. There are not enough data points to say for sure. Could be my imagination but it also appears to be less smooth. If possible, please add additional values of $S$.

---

> > > > > ### Author Response · Authors · 2021-08-29
> > > > > **We thank reviewer for your further comments**
> > > > >
> > > > > As suggested by the reviewer, we have evaluated additional values of S over the range {1, 5, 10, 15, 20} on two image datasets: SVHN and CIFAR10. [Figure 7](https://www.dropbox.com/s/8kw7j7kgylygzn7/figure_7.pdf?dl=0) shows that the validation errors decrease faster for larger values of S, but reach a similar level for all values of S. It indicates that the validation performance doesn't degrade as the sample size S increases.

---

### Official Review · Reviewer_udxa · 2021-07-12

**Rating:** 7
**Confidence:** 3

**Summary:**

The authors propose to perform joint inference over the depth and effective width of a deep neural network. The depth is modelled as a stochastic process (in particular a Beta process), which allows for the depth to be infinite in theory. Given a fixed maximum width, the effective width is determined using a conjugate Bernoulli distribution which models the dropout probability for each neuron in each layer.

**Limitations And Societal Impact:**

The authors do not discuss limitations. One example would be that the proposed method doesn't easily apply to modern neural network architectures such as VGG or ResNet where the layers at different depths are not the same size and which include things like pooling, residual connections and a mix of convolutional and dense layers.

The authors also do not discuss the potential negative impact of their work. One thing that could be mentioned is the consequences of miscalibrated predictions, particularly for safety-critical applications.

**Main Review:**

## Pros

### Thorough experimental evaluation

The authors have provided a wide range of benchmarks comparing their proposed method against numerous baselines from the literature. It was great to see a mix of experiments including toy data, small scale regression, image classification, and continual learning. The experiments shown in figures 4 and 6 were both interesting and worthwhile inclusions. The additional experiments in the appendix were also appreciated.

However, there is also room for improvement here. The paper was missing:

* An ablation study to disentangle the effects of inference over depth and dropout.

* Deep ensembles as a baseline. This is very important as a standard point of reference in the Bayesian deep learning literature and it is simple to implement.

* Some out-of-distribution (rather than distribution shift) benchmarks. For example out- vs in-distribution classification using entropy or the max logit for CIFAR10 vs SVHN, CIFAR10 vs CIFAR100, and MNIST vs FashionMNIST. Similarly for OOD rejection classification. See Antoran et al. (2020) for descriptions of these.

* A wider range of toy data experiments in order to build intuition for the model's behaviour.

* Dropout as a baseline for all of the experiments (given its strong performance on the UCI regression benchmark).

### Strong empirical performance

Not only was the experimental evaluation thorough but it also showed that the authors proposed method works well!

### Theoretical soundness
As far as I can tell the proposed method is sound. However, the caveat is that I am not an expert in Bayesian non-parametric models.

### Novelty

Other papers have done inference over depth (e.g. Antoran et al. (2020)) and over depth + width (e.g. Nalisnick et al. (2019), Dikov and Bayer (2019)). The main novelty of this paper is that there is no fixed upper limit on the depth in theory. However, as with any Bayesian non-parametric model in practice, we need to apply truncation. As a result, this method does have a maximum depth. However, as shown in figure 12, the method allows for a large truncation level without a large increase in train/test time, which is *not* the case in previous depth inference works.

## Cons

### Clarity

The paper was not clearly written. In many places, there were typos and grammatical mistakes. I would urge the authors to use a tool like Grammarly to help fix these issues.

### Plagiarism concerns

This is my biggest issue with this paper. Numerous of the sentences were taken from Antoran et al. (2020) and modified in small/trivial ways. Similarly, several of the figures and tables were adapted with only small changes. These issues are mostly in the experiment descriptions. I am not suggesting that the main idea or contributions of this paper are plagiarised, nor am I suggesting that there was any malice or ill-intent from the authors. However, I do not believe that the paper can be published in this state.

A (non-exhaustive) list of such issues:

* "All methods’ accuracy degrades quickly for rotations larger than 30◦, but our method is the least overconfident" (this paper)
vs
"Although all methods perform well on the original test-set, their accuracy degrades quickly for rotations larger than 30°. Here, DUNs and S-ResNets differentiate themselves by being the least overconfident" (Antoran et al.).

* "... we train the methods on CIFAR10 and evaluate them on data subject to 16 different corruptions with 5 levels of intensity each" (this paper)
vs
"we train models on CIFAR10 and evaluate them on data subject to 16 different corruptions with 5 levels of intensity each"  (Antoran et al.).

* "We perform Bayesian Optimisation and Hyperband (BOHB) to determine the best configurations including backbone-structure depths and hyperparameter settings for each method" (this paper)
vs
"We select all hyperparameters, including NN depth, using Bayesian optimisation with HyperBand" (Antoran et al.).

* "... employ BOHB that combines the strengths of BO (strong final performance) and Hyperband (scalability, flexibility, and robustness). In particular, we use the HpBandSter implementation of Bayesian Optimization and Hyperband (BOHB) (https://github.com/automl/HpBandSter)." (this paper)
vs
"... we use Bayesian Optimisation and Hyperband (BOHB). This method, as the name suggests, combines BO with Hyperband, a bandit based HPO method. BOHB has the strengths of both BO (strong final performance) and Hyperband (scalability, flexibility, and robustness). In particular, we use the HpBandSter implementation of BOHB: https://github.com/automl/HpBandSter. " (Antoran et al.).

* "We select the best configurations for each method on the standard splits of UCI regression benchmark datasets using the settings, as shown in Table 1. We find these settings to be sufficiently large to ensure all methods’ convergence." (this paper)
vs
"We run BOHB for each dataset and split for 20 iterations using the same settings, shown in Table 2. We find these values to be sufficiently large to ensure all methods’ convergence." (Antoran et al.).

* Compare figure 3 of this paper to figure 5 of Antoran et al. – it looks to me like the authors of this paper used the code of Antoran et al without proper attribution.

* Similarly, compare tables 1-4 in this paper with figures 2-5 in Antoran et al.

* Compare Appendix D in this paper with Appendix D in Antoran et al. This is similarly filled with minor rewrites and most likely uses the same LaTeX code for the formulae.

## Summary

Overall I thought that the paper was interesting, theoretically sound and provided a strong performance on a range of benchmark tasks vs several baselines. However, due to clarity issues, some missing experimental results, and the liberal usage of content from Antoran et al., I can't recommend acceptance at this point. Should the authors
1) add OOD experiments, the deep ensembles baseline, and ideally the requested ablation study, as well as
2) satisfactorily address the offending sections of the paper by rewriting and attributing to Antoran et al. where appropriate,
I would be happy to increase my score to a 7.

## Other minor suggestions

Show test LL rather than accuracy for fig 4 since this measure both uncertainty calibration as well as the goodness of fit. (At least add the same plot with LL to the appendix.)

Show results for M={16,32} in fig 6. This will show the behaviour better since we have saturation after M=64.

## Post-rebuttal update

Given that all of my concerns have been addressed by the authors, I have increased my score to a 7.

**Time Spent Reviewing:**

9

---

> ### Author Response · Authors · 2021-08-10
> **We thank reviewer for your essential and constructive comments. Our detailed response is as below.**
>
> - **Plagiarism concern**  we take the reviewer’s plagiarism concern very seriously. We will re-write the corresponding experiment descriptions, and make proper attributions for the tools we used to generate the figures. For Table 1-4, we consistently use the same setting for BOHB and the derived optimized configurations as in (Antoran et al) for all methods in order to make a fair and rigorous performance comparison. We will clarify them in the camera-ready version.
>
>
>     For example, We will rewrite ``All methods’ accuracy degrades quickly for rotations larger than 30 degrees, but our method is the least overconfident`` as  ``Figure 5(a) shows that the uncertainty calibration of our method is significantly more robust to the data with large angle rotations (greater than 30 degrees) than other methods, and it achieves the performance with lowest expected confidence estimates measured with the formula in (Antoran et. al.).``, and ``... we train the methods on CIFAR10 and evaluate them on data subject to 16 different corruptions with 5 levels of intensity each`` as ``To make a fair and rigorous comparison as in (Antoran et al.), we also evaluate the corruption robustness of all methods on the same CIFAR10 dataset with 16 types of algorithmically generated corruptions. Each type of corruption has five levels of severity.``
>
>     We plotted Figure 3 using the codes of Antoran et al. We will add a statement in the caption ``The plots are drawn with the codes of (Antoran et al).`` to make proper attribution.
>
>     For the same settings of BOHB and the optimized configurations in the Appendix Table 1-4, we will add a statement ``We follow (Antoran et al) for the setting of BOHB, and generate the optimized configurations for all methods for a fair comparison.``
>
>     For Appendix D, We will re-write the descriptions, and add a statement ``We adopt the formulae as defined in (Antoran et al.) for uncertainty calibration evaluations.``
>
> - **Deep Ensemble as baseline** We compare the performance of our method with the deep ensemble results reported in [Lakshminarayanan et. al 2017] as in Table 5.  We also evaluate the deep ensembles' uncertainty calibration in [Figure 2](https://www.dropbox.com/s/cexec6drprh1fjz/figure_2.pdf?dl=0), OOD detection in Table 6, and [Figure 1](https://www.dropbox.com/s/dxvbm8sywpxuumq/figure_1.pdf?dl=0). Overall, deep ensembles is the least efficient method with comparable performance to others. However, [Figure 2](https://www.dropbox.com/s/cexec6drprh1fjz/figure_2.pdf?dl=0) and [Figure 1](https://www.dropbox.com/s/dxvbm8sywpxuumq/figure_1.pdf?dl=0) show deep ensembles performs better for uncertainty calibration and OOD detection. We will incorporate these deep ensembles into the camera-ready version.
>
> **Table 5:** Performance comparison of our method and Deep Ensembles on regression datasets
>
> | Dataset  |        Log-likelihood        |                         |          RMSE           |                    |
> | :------- | :--------------------------: | :---------------------: | :---------------------: | :----------------: |
> |          |           **Ours**           |   **Deep Ensembles**    |        **Ours**         | **Deep Ensembles** |
> | Boston   |   $\mathbf{-2.36\pm0.22}$    |     $-2.41\pm0.25$      | $\mathbf{2.61\pm0.68}$  |   $3.28\pm1.00$    |
> | Concrete |   $\mathbf{-2.95\pm0.06}$    |     $-3.06\pm0.18$      | $\mathbf{4.58\pm0.36}$  |   $6.03\pm0.58$    |
> | Energy   |   $\mathbf{-0.79\pm0.07}$    |     $-1.38\pm0.22$      | $\mathbf{0.49\pm0.06}$ ​ |   $2.09\pm0.29$​    |
> | Kin8nm   | $\ \ \ \mathbf{1.24\pm0.02}$ |   $\ \ \ 1.20\pm0.02$   | $\mathbf{0.07\pm0.01}$  |   $0.09\pm0.00$    |
> | Naval    | $\ \ \ \mathbf{5.79\pm0.18}$ |   $\ \ \ 5.63\pm0.05$   |      $0.00\pm0.00$      |   $0.00\pm0.00$    |
> | Power    |   $\mathbf{-2.70\pm0.18}$    |     $-2.79\pm0.04$      | $\mathbf{3.61\pm0.19}$  |   $4.11\pm0.17$    |
> | Protein  |   $\mathbf{-2.57\pm0.01}$    |     $-2.83\pm0.02$      | $\mathbf{3.37\pm0.03}$  |   $4.71\pm0.06$    |
> | Wine     |        $-0.98\pm0.09$        | $\mathbf{-0.94\pm0.12}$ | $\mathbf{0.60\pm0.05}$  |   $0.64\pm0.04$    |
> | Yacht    |   $\mathbf{-0.98\pm0.11}$    |     $-1.18\pm0.21$      | $\mathbf{0.71\pm0.24}$  |   $1.58\pm0.48$    |
>
> - **Out-of-distribution benchmarks** We conduct additional evaluations on two in-distribution and out-distribution test sets, as in Table 6. The overall accuracies of our method are competitive with other baselines including deep ensembles. First, we train the methods on CIFAR10 (and MNIST) dataset and evaluate them on the in-distribution CIFAR10 (MNIST) test set and out-distribution SVHN (FashionMNIST) test set. Table 6 shows the performance of all methods based on predictive entropy. Second, we evaluate the methods on an OOD rejection task. We sort the samples based on their entropy and reject different proportions of the samples with large entropy. The model then makes predictions on the rest of the samples, and the accuracy is reported in [Figure 1](https://www.dropbox.com/s/dxvbm8sywpxuumq/figure_1.pdf?dl=0). As the reviewer suggested, we will incorporate these results into the Appendix for the camera-ready version.
>
>     **Table 6:** AUROC for in- vs out-distribution classification with predictive entropy.
>
> | Methods          | MNIST vs FashionMNIST | CIFAR10 vs SVHN |
> | :--------------- | :-------------------: | :-------------: |
> | Our method       |    $0.968\pm0.028$    | $0.869\pm0.014$ |
> | DUN              |    $0.847\pm0.014$    | $0.838\pm0.033$ |
> | Stochastic Depth |    $0.912\pm0.001$    | $0.863\pm0.018$ |
> | Concrete Dropout |    $0.951\pm0.001$    | $0.855\pm0.009$ |
> | Deep Ensembles   |    $0.991\pm0.002$    | $0.909\pm0.001$ |
> | Dropout          |    $0.956\pm0.015$    | $0.899\pm0.004$ |
>
> - **Dropout as baseline** We reported dropout performance on the toy and UCI data sets. We include dropout in the evaluations of uncertainty calibration in [Figure 2](https://www.dropbox.com/s/cexec6drprh1fjz/figure_2.pdf?dl=0). We will incorporate it into the camera-ready version.
>
> - **Additional toy experiments** As suggested by the reviewer, we conduct an additional toy experiment, as shown in  [Figure 3](https://www.dropbox.com/s/pqdusmmfr1sg0of/figure_3.pdf?dl=0). We generate 2d data points from a 2-armed spiral function with varying degrees of rotations  ($180^{\circ}, 360^{\circ}$, and $720^{\circ}$) while keeping the number of training data points fixed to $200$. For inference, we set hyper-parameters $\alpha=1.1$ and $\beta=2$ with maximum number of neurons in each hidden layer as $M = 20$. Other experimental settings are the same as in Section C.1. As in  [Figure 3](https://www.dropbox.com/s/pqdusmmfr1sg0of/figure_3.pdf?dl=0), our model enables the neural architecture to scale up by activating more layers and neurons to adapt to the increasing complexity of the underlying data generation process. In particular, it dynamically balances the depth and width of the neural architecture. We will add the experiment to the Appendix.
>
> - **Clarity**  We will proofread and spell-check the draft, and improve its clarity.
>
> - **Limitations about the choice of architectures** We will address the limitations and societal impact of our method as suggested by the reviewer in the camera-ready version.

---

> > ### Comment · Reviewer_udxa · 2021-08-19
> > **Thank you for the detailed response and additional results**
> >
> > Thanks for providing so many additional experimental results. I think that the wide range of additional results provided here (and in response to the other reviews) will really strengthen the paper.
> >
> > I also appreciate that the plagiarism concerns have been taken seriously. The examples of changes that will be made are good. In fact, the citations of Antoran et al., could even be toned down slightly from the proposal. For example, you could simply mention that you used code from Antoran et al. to make some of the plots in your appendix rather than in the captions themselves.
> >
> > Given that all of my experimental suggestions have been carried out and that the plagiarism issue seems to be resolved, I am happy to increase my score to 7.
> >
> > (Minor, but since it wasn't mentioned in the response, please do show results for M={16,32} in fig 6!)

---

> > > ### Author Response · Authors · 2021-08-24
> > > **We thank the reviewer for your further comments**
> > >
> > > We will make sure the camera-ready version will reflect the changes to address the plagiarism concerns. We will also tone down the citation of Antoran et al. in the way suggested by the reviewer.
> > >
> > > As suggested by the reviewer, we added the results for M={16, 32} in [Figure 6](https://www.dropbox.com/s/feir6vt9fzbaxs6/figure_6.pdf?dl=0). We will incorporate this result in our camera-ready version.

---

### Official Review · Reviewer_tkqv · 2021-07-16

**Rating:** 6
**Confidence:** 4

**Summary:**

The paper proposes a comprehensive but effective method to simultaneously perform inference for the NN depth and width. As the idea might not be new, the execution of the method is solid. The methodology is also linked to Bayesian information criterion with a theorem.

**Limitations And Societal Impact:**

Yes.

**Main Review:**

Originality:
The structure selection has been studies by a few related works. This work models the network depth with Beta process while the Bernoulli variables indicate the if a node is kept or dropped. With this modeling, it is easy to control the network shape with the distribution parameter - alpha and beta, which is smart. It is interesting to see behaviors of network shape under different (alpha, beta) setups. The inference is done with standard stochastic variational inference.

Quality, clarity, and significance:
1. The paper is well-written and easy to read. The graphs are clear and intuitive.
2. The major doubt comes from the experiment section.

-As the network architecture (depth and width) is considered, why there's no results on efficiency reported?

-The test error rate on Cifar10 is over 10%, which is not a competitive results. With standard VGG network, it could easily achieve less than 10%. What is the network used for Cifar10?

-Only small scale datasets are used. Does the method scale to large datasets? For example, TinyImagenet might be a proper choice for Bayesian deep learning.

-How is the performance of detecting out-of-domain data?

-How is the performance compared with deep ensembles? Or other Bayesian method related to sparsity and model selection? For example:

Louizos et al, Learning sparse neural networks through l 0 regularization, ICML 2018.
Dai et al., Compressing Neural Networks using the Variational Information Bottleneck, ICML 2018.
Cui et al, Bayesian Nested Neural Networks for Uncertainty Calibration and Adaptive Compression, CVPR 2021.

**Time Spent Reviewing:**

6

---

> ### Author Response · Authors · 2021-08-10
> **We thank reviewer for your essential and constructive comments. Our detailed response is as below.**
>
> - **Efficiency** We provided some analysis of time complexity in Appendix Line 36-38. Table 2 shows the inference and prediction times of all methods. In particular, for deep ensembles, we evaluate the neural network with depth $\{5, 10, 15, 20\}$, and 5 neural networks are trained for each case to determine the best configuration. Inference only needs to be done once for our method vs multiple times for deep ensembles. For training a dropout neural network that has depth $L$ and width $M$, The time complexity is $O(NBLM^2)$ with $N$ training examples and $B$ epochs, given fixed batch size. Let $T$ be the time cost of a single NN forward pass, with $S$ samples our method is linearly scalable as $ST$. Our method is no more than an order of magnitude faster than other dropout variants. Since the stick-breaking construction of the beta process induces that $\pi_k$, the probability of seeing activated neurons in hidden layers, decreases exponentially with truncation level $K$. With proper thresholding, only a relatively small number of hidden layers are sampled in training. For prediction, our method is about as fast as other dropout variants with a small additional memory overhead. Deep ensembles require multiple forward passes which leads to slower prediction speed. As the reviewer suggested, we'll clarify this in the camera-ready version.
>
>     **Table 2:** The inference and prediction times of all methods with batch size 256. We measure the CPU times (in seconds) for MNIST and CIFAR-10 data sets on a single NVIDIA RTX 2080Ti GPU. For deep ensembles, we evaluate the neural network with depth $\{5, 10, 15, 20\}$, and 5 neural networks are averaged for each case to determine the best configuration. For inference, we report the average time per epoch. The prediction time is what it takes to make predictions for all test samples.
>     |     Methods      | Inference |             | Prediction |             |
>     | :--------------: | :-------: | :---------: | :--------: | :---------: |
>     |                  | **MNIST** | **CIFAR10** | **MNIST**  | **CIFAR10** |
>     |    Our method    | $18.314$  |  $27.232$   |  $1.016$   |   $1.016$   |
>     |     Dropout      | $10.128$  |  $16.717$   |  $0.469$   |   $1.133$   |
>     |       DUN        | $14.978$  |  $12.142$   |  $0.820$   |   $0.938$   |
>     |  Deep Ensembles  | $50.642$  |  $83.583$   |   $5.01$   |   $8.281$   |
>     | Stochastic Depth | $22.538$  |  $41.125$   |  $1.406$   |   $1.953$   |
>     | Concrete Dropout | $14.325$  |  $15.423$   |  $ 0.938$​  |   $1.25$    |
>     |   IBP Dropout    | $37.368$  |  $37.063$   |  $1.273$   |   $1.523$   |
>
> - **Test error rate on CIFAR10** The neural networks structures are described in Appendix Section C.3. We employ a chain-structured space with an unbounded number of hidden layers. For the input block, we use $5 \times 5$ kernel convolutional layer combined with Leaky ReLU activation functions and $2 \times 2$ max pooling. The output block is composed of a global averaging layer followed by a fully connected residual block and a linear layer. We apply the softmax operation to the output activation functions. Each hidden layer consists of a $3 \times 3$​ convolutional layer followed by a Leaky ReLU activation and Batch Normalization. These are also the basic setup for other competing methods. As in the related works section, multiple studies show that very deep networks tend to have a calibration problem with high confidence on incorrect predictions. We propose a method to address the problem. It is part of our research interest to extend our method to modern network architectures such as VGG. We will clarify this limitation in the camera-ready version.
>
> - **Small scale datasets** For fair and rigorous comparison, the benchmark datasets we evaluate in this paper are the baselines widely used for all competing dropout variants and structure selection methods. Based on the above complexity analysis, it is readily to extend our method to even larger data sets.
>
> - **OOD detection** We conduct additional evaluations on two in-distribution and out-distribution test sets, as in Table 3. The overall accuracies of our method are competitive with other baselines including deep ensembles. First, we train the methods on CIFAR10 (and MNIST) dataset and evaluate them on the in-distribution CIFAR10 (MNIST) test set and out-distribution SVHN (FashionMNIST) test set. Table 3 shows the performance of all methods based on the predictive entropy. Second, we evaluate the methods on an OOD rejection task. We sort the samples based on their entropy and reject different proportions of the samples with large entropy. The model then makes predictions on the rest of the samples, and the accuracy is reported in [Figure 1](https://www.dropbox.com/s/dxvbm8sywpxuumq/figure_1.pdf?dl=0). As the reviewer suggested, we will incorporate these results into the Appendix for the camera-ready version.
> **Table 3:** AUROC for in- vs out-distribution classification with predictive entropy.
>
> | Methods          | MNIST vs FashionMNIST | CIFAR10 vs SVHN |
> | :--------------- | :-------------------: | :-------------: |
> | Our method       |    $0.968\pm0.028$    | $0.869\pm0.014$ |
> | DUN              |    $0.847\pm0.014$    | $0.838\pm0.033$ |
> | Stochastic Depth |    $0.912\pm0.001$    | $0.863\pm0.018$ |
> | Concrete Dropout |    $0.951\pm0.001$    | $0.855\pm0.009$ |
> | Deep Ensembles   |    $0.991\pm0.002$    | $0.909\pm0.001$ |
> | Dropout          |    $0.956\pm0.015$    | $0.899\pm0.004$ |
>
> ​
> - **Deep Ensembles**  We evaluate the performance of deep ensembles on the UCI data sets and compare it with our method in Table 4. We also analyze and measure the computation time of deep ensembles in Table 2, and evaluate its uncertainty calibration in [Figure 2](https://www.dropbox.com/s/cexec6drprh1fjz/figure_2.pdf?dl=0), and its OOD detection in Table 3, and  [Figure 1](https://www.dropbox.com/s/dxvbm8sywpxuumq/figure_1.pdf?dl=0). Deep ensembles are the least efficient method with comparable predictive and OOD detection performance to all other methods, as shown in Table 2, Table 3, and [Figure 1](https://www.dropbox.com/s/dxvbm8sywpxuumq/figure_1.pdf?dl=0). However, [Figure 2](https://www.dropbox.com/s/cexec6drprh1fjz/figure_2.pdf?dl=0) and [Figure 1](https://www.dropbox.com/s/dxvbm8sywpxuumq/figure_1.pdf?dl=0) show that deep ensembles perform better for uncertainty calibration and OOD detection. We will incorporate the results of deep ensembles into the camera-ready version.
>
> **Table 4:** Performance comparison of our method and Deep Ensembles on regression datasets
>
> | Dataset  |        Log-likelihood        |                         |          RMSE          |                    |
> | :------- | :--------------------------: | :---------------------: | :--------------------: | :----------------: |
> |          |           **Ours**           |   **Deep Ensembles**    |        **Ours**        | **Deep Ensembles** |
> | Boston   |   $\mathbf{-2.36\pm0.22}$    |     $-2.41\pm0.25$      | $\mathbf{2.61\pm0.68}$ |   $3.28\pm1.00$    |
> | Concrete |   $\mathbf{-2.95\pm0.06}$    |     $-3.06\pm0.18$      | $\mathbf{4.58\pm0.36}$ |   $6.03\pm0.58$    |
> | Energy   |   $\mathbf{-0.79\pm0.07}$    |     $-1.38\pm0.22$      | $\mathbf{0.49\pm0.06}$​ |   $2.09\pm0.29$​    |
> | Kin8nm   | $\ \ \ \mathbf{1.24\pm0.02}$ |   $\ \ \ 1.20\pm0.02$   | $\mathbf{0.07\pm0.01}$ |   $0.09\pm0.00$    |
> | Naval    | $\ \ \ \mathbf{5.79\pm0.18}$ |   $\ \ \ 5.63\pm0.05$   |     $0.00\pm0.00$      |   $0.00\pm0.00$    |
> | Power    |   $\mathbf{-2.70\pm0.18}$    |     $-2.79\pm0.04$      | $\mathbf{3.61\pm0.19}$ |   $4.11\pm0.17$    |
> | Protein  |   $\mathbf{-2.57\pm0.01}$    |     $-2.83\pm0.02$      | $\mathbf{3.37\pm0.03}$ |   $4.71\pm0.06$    |
> | Wine     |        $-0.98\pm0.09$        | $\mathbf{-0.94\pm0.12}$ | $\mathbf{0.60\pm0.05}$ |   $0.64\pm0.04$    |
> | Yacht    |   $\mathbf{-0.98\pm0.11}$    |     $-1.18\pm0.21$      | $\mathbf{0.71\pm0.24}$ |   $1.58\pm0.48$    |

---

### Official Review · Reviewer_fgUw · 2021-07-18

**Rating:** 6
**Confidence:** 4

**Summary:**

The authors propose a framework that performs dropout and infers the number of layers jointly for neural network.

**Limitations And Societal Impact:**

I do not think potentially negative societal impacts is applicable to this work.

**Main Review:**

I believe that learning neural network architecture, uncertainty quantification, and dropout regularization in neural networks are all interesting and important topics of research.

The paper is written in a clear, accessible and well organized way.

I think the main pro going for this paper is the experiments, which shows that this approach captures uncertainty in a reasonable and intuitive way that exceeds or is in general comparable to other methods. The experimental performance is the sole reason I am giving it an above acceptance threshold rating.

Several comments below that I think if addressed, could make the paper stronger/clearer:
1. Computation: there is little mention of computational time or scalability or big Oh running times in the main paper. I think this is a key factor. In my experience dealing with bayesian neural networks, even with variational inference computational and convergence is sometimes an issue, and I think the authors should analyze and compare the computational complexity/actual running time of their methods to other methods more explicitly. My experience working with anything involving indian buffet processes is that it often messes things up/slows things down, so I am curious to see what the author's experience with this is in their work.
2. Motivation: I find the motivation of the setup, or the lack thereof, slightly unsatisfying.  It seems like the authors just decided to put the most convenient prior they could think of on the layer number, conveniently pick some truncation threshold, and then just applied variational inference and called it a day. I do not blame the authors for this, since choice of priors and computational issues are difficult open problems and this happens in many a Bayesian papers, and but if they have any rationale or deeper reasons other than convenience to pick these priors, I would like to hear it.

In particular, one motivation that the authors mentioned is that existing methods can often only reduce depth of a fixed architecture, but in the case of this work, I suspect that if one does the truncation, and set the hyperparameters in any reasonable kind of way, with very high probability the posterior is going to allocate most probability on networks with number of layers below some (potentially big) constant number, and then basically this method is also a priori putting a lot of mass onto networks with number of layers below a certain threshold, which kind of defeats a small part of its purpose.

**Time Spent Reviewing:**

2 hours

---

> ### Author Response · Authors · 2021-08-10
> **We thank reviewer for your essential and constructive comments. Our detailed response is as below.**
>
> - **Computation** some complexity analysis is in the Appendix Line 36-38. For training a dropout neural network that has depth $L$​ and width $M$​, The time complexity is $O(NBLM^2)$​ with $N$​ training examples and $B$​ epochs, given fixed batch size. Let $T$​ be the time cost of a single NN forward pass, with $S$​ samples our method is linearly scalable as $ST$​. Our method is no more than an order of magnitude faster than other dropout variants as shown in Table 1. Since the stick-breaking construction of beta process induces that $\pi_k$​, the probability of seeing activated neurons in hidden layers, decreases exponentially with the truncation level $K$​. With proper thresholding, only a relatively small number of hidden layers are sampled in training. On the contrary, the adaptive sparsification method computes an Indian buffet process per hidden layer. This slows it down in training. For prediction, our method is about as fast as other dropout variants with a small additional memory overhead. Deep ensembles require multiple forward passes which leads to slower prediction speed. We'll clarify these results in the camera-ready version, as the reviewer suggested.
>
>      **Table 1:** The inference and prediction times of all methods with batch size 256. We measure the CPU times (in seconds) for MNIST and CIFAR-10 data sets on a single NVIDIA RTX 2080Ti GPU. For deep ensembles, we evaluate the neural network with depth $\{5, 10, 15, 20\}$, and 5 neural networks are averaged for each case to determine the best configuration. For inference, we report the average time per epoch. The prediction time is what it takes to make predictions for all test samples.
>
>     |     Methods      | Inference |             | Prediction |             |
>     | :--------------: | :-------: | :---------: | :--------: | :---------: |
>     |                  | **MNIST** | **CIFAR10** | **MNIST**  | **CIFAR10** |
>     |    Our method    | $18.314$  |  $27.232$   |  $1.016$   |   $1.016$   |
>     |     Dropout      | $10.128$  |  $16.717$   |  $0.469$   |   $1.133$   |
>     |       DUN        | $14.978$  |  $12.142$   |  $0.820$   |   $0.938$   |
>     |  Deep Ensembles  | $50.642$  |  $83.583$   |   $5.01$   |   $8.281$   |
>     | Stochastic Depth | $22.538$  |  $41.125$   |  $1.406$   |   $1.953$   |
>     | Concrete Dropout | $14.325$  |  $15.423$   |  $ 0.938$  |   $1.25$    |
>     |   IBP Dropout    | $37.368$  |  $37.063$   |  $1.273$   |   $1.523$   |
>
> - **Motivation** The main motivation for us to adopt the *beta process* to model the number of hidden layers is because it is a complete random measure. It assumes the activation probabilities of hidden layers are independent, each of which is in [0,1]. For comparison, The *Dirichlet process* has a constraint that the activation probabilities of all the hidden layers have to sum to 1. It implies that if there are some hidden layers with higher activations, the others have to be less activated, which is counterintuitive. The *Beta process* is derived from a *Poisson process* which is another complete random measure, and they share similar characteristics. *Indian buffet process* is a marginalized version of *beta-Bernoulli processes* over probabilities, and it usually models a fixed number of objects with infinite features. With an efficient inference algorithm, we consider the *beta process* as a sensible choice. For our future research, we will also investigate some recent stochastic processes (e.g., random function priors). We will clarify it in the camera-ready version.
>
> - **Truncation** we agree with the reviewer. How to relax the constraint of truncation is also part of our research interest. One idea is to conduct probabilistic inference for truncation level by treating it as a random variable.

---

### Decision · Program_Chairs · 2021-09-27

**Decision:**

Accept (Poster)

**Comment:**

The submission proposes a model and variational inference scheme to learn the depth and width of neural networks simultaneously. All reviewers were impressed with the strong empirical performance of the proposed method compared to multiple baselines in a suite of tasks. One reviewer raised a concern that some text in the experiments were copied directly from Antoran et al. (2020), which the authors promised to rewrite. Overall, a good paper and should be of interest to the community.